# Bim escapes displacement by BH3-mimetic anti-cancer drugs by double-bolt locking both Bcl-XL and Bcl-2

Qian Liu[1], Elizabeth J Osterlund[1,2], Xiaoke Chi[3], Justin Pogmore[2], Brian Leber[4], David William Andrews[2]*

[1]Biological Sciences, Sunnybrook Research Institute, Toronto, Canada; [2]Department of Biochemistry, University of Toronto, Toronto, Canada; [3]Department of Biochemistry and Biomedical Sciences, McMaster University, Hamilton, Canada; [4]Department of Medicine, McMaster University, Hamilton, Canada

**Abstract** Tumor initiation, progression and resistance to chemotherapy rely on cancer cells bypassing programmed cell death by apoptosis. We report that unlike other pro-apoptotic proteins, Bim contains two distinct binding sites for the anti-apoptotic proteins Bcl-XL and Bcl-2. These include the BH3 sequence shared with other pro-apoptotic proteins and an unexpected sequence located near the Bim carboxyl-terminus (residues 181–192). Using automated Fluorescence Lifetime Imaging Microscopy - Fluorescence Resonance Energy Transfer (FLIM-FRET) we show that the two binding interfaces enable Bim to double-bolt lock Bcl-XL and Bcl-2 in complexes resistant to displacement by BH3-mimetic drugs currently in use or being evaluated for cancer therapy. Quantifying in live cells the contributions of individual amino acids revealed that residue L185 previously thought involved in binding Bim to membranes, instead contributes to binding to anti-apoptotic proteins. This double-bolt lock mechanism has profound implications for the utility of BH3-mimetics as drugs.

DOI: https://doi.org/10.7554/eLife.37689.001

*For correspondence:
david.andrews@sri.utoronto.ca

**Competing interests:** The authors declare that no competing interests exist.

## Introduction

A major function of apoptosis is to inhibit tumor initiation and progression while its inhibition can result in cancer cell survival and resistance to chemotherapy (*Kirkin et al., 2004*). Bcl-2 family proteins regulate intrinsic apoptosis and mitochondrial integrity through direct physical interactions between the anti-apoptotic proteins (Bcl-2, Bcl-XL and Mcl-1), the pro-apoptotic proteins (Bax, Bak) and the BH3-proteins (Bad, Bid and Bim) (*Liu et al., 2013*; *Shamas-Din et al., 2013*; *Youle and Strasser, 2008*). In cancer cells, pro-apoptotic BH3 proteins and activated Bax and Bak are sequestered by anti-apoptotic proteins, thereby the cell is protected at the same time as it is primed for death. Such cells can be efficiently killed by BH3-mimetic drugs that release bound pro-apoptotic proteins, as shown by the clinical success of the Bcl-2 specific BH3-mimetics venetoclax (ABT-199) and navitoclax (ABT-263).

It is commonly accepted that BH3 proteins bind to anti-apoptotic proteins exclusively through a conserved BH3 region (*Chen et al., 2005*). ABT-263 is an orally available inhibitor with nanomolar affinity for the BH3-binding sites in Bcl-XL and Bcl-2 that is currently undergoing clinical trials in humans. In vitro ABT-263 displaces BH3 peptides derived from different BH3-proteins including Bim (*Tse et al., 2008*) from Bcl-2 and Bcl-XL. However a discrepancy between this data and the activities of the drugs in live cells was observed independently in three studies. 1) Increased expression of Bcl-XL in human non-Hodgkin lymphomas led to resistance to ABT-737 (an analogue of ABT-263) despite enhanced Bim expression (*Mérino et al., 2012*). 2) Fluorescent Lifetime Imaging Microscopy

**eLife digest** The body can get rid of cells that are unnecessary, infected or damaged by instructing them to go through a process known as apoptosis, which results in the cell killing itself. These orders are given in the form of pro-death molecules, such as the BH3 proteins, which kick start apoptosis.

In order to survive and multiply, cancer cells can increase the levels of anti-death proteins which bind and 'trap' BH3 proteins, preventing them from triggering apoptosis. At the molecular level, the process involves the anti-death proteins recognizing and attaching to a specific 'BH3 sequence' in the pro-death signals.

A new type of anti-cancer treatment works by tricking the anti-death proteins into binding a drug rather than BH3 proteins, which are then free to induce apoptosis. These decoy drugs mimic the BH3 sequences that the anti-death proteins recognize and attach to. However, studies have shown that the BH3-protein Bim stays bound to the body's anti-death proteins rather than being displaced by the drugs. In patients, this means that the cancer cells may resist and survive the treatment.

Here, Liu et al. try to understand why Bim keeps on attaching to anti-death proteins by developing an advanced microscopy technique to dissect the interactions between the two types of molecules. This revealed that Bim has a second sequence for binding to anti-death proteins, which is located far away from the 'normal' BH3 sequence. Together, the two sequences allow Bim to double-bolt lock to anti-death proteins, which explains why it cannot be displaced by drugs that mimic only the BH3 sequence.

In the future, the new binding sequence may serve as a target for anti-cancer drugs. The microscopy technique developed by Liu et al. can also be used to study other pairs of interacting proteins.

DOI: https://doi.org/10.7554/eLife.37689.002

(FLIM) – Förster Resonance Energy Transfer (FRET) demonstrated that ABT-263 displaced full-length Bad and tBid (activated Bid protein) but not the major isoforms of Bim (BimEL, BimL and BimS) from binding to Bcl-XL and Bcl-2 (*Aranovich et al., 2012*; *Liu et al., 2012*). 3) ABT-263 resistant binding of Bim to Bcl-XL was observed using bioluminescence resonance energy transfer (*Pécot et al., 2016*). However, the molecular mechanism of resistance is not well established.

Among the Bcl-2 family, Bim is the only BH3-protein that binds all anti-apoptotic proteins with high affinities and that can directly activate Bax and Bak to initiate apoptosis (*Chen et al., 2005*; *Letai et al., 2002*). Therefore the surprising lack of activity for BH3 mimetics on displacement of Bim could significantly reduce their efficacy in some cancer cells (*Pécot et al., 2016*). Understanding this mechanism of resistance may also suggest ways to release specific sequestered BH3-proteins from Bcl-XL and Bcl-2 to kill cancer cells more selectively than broadly inhibiting anti-apoptotic proteins (*Goldsmith et al., 2006*; *Ni Chonghaile and Letai, 2008*).

Here we identify the specific residues in Bim responsible for binding to Bcl-XL and Bcl-2 and that confer resistance of the complexes to BH3-mimetic drugs. We discovered an unanticipated interaction between the C-terminal sequence (CTS) of Bim and both Bcl-XL and Bcl-2. Our data further demonstrate dual yet independent roles for the Bim CTS in binding to Bcl-XL and to membranes. Together the two anti-apoptotic protein binding sequences in Bim confer resistance of complexes to the dual Bcl-2, Bcl-XL inhibitor ABT-263 and to the new generation Bcl-XL inhibitors in clinical development. Because the CTS of Bim also functions to activate the pro-apoptotic proteins Bax and Bak, (*Chi et al., 2019*) our results suggest that BH3-mimetics may have unpredictable results in cells depending on the Bcl-2 family proteins expressed and that targeting both binding sites on Bcl-XL and/or Bcl-2 may be required to kill some cancer cells that have bypassed apoptosis by inhibiting Bim.

## Results

### ABT-263 displaces Bad and tBid but not Bim from Bcl-XL and Bcl-2

New instrumentation for automated time correlated single photon counting (TCSPC) from two fluorescence proteins simultaneously enabled automation of FLIM-FRET measurements to evaluate the effects of drugs on the binding of Bcl-2 family proteins fused to mCerulean3 (mCer3) and Venus fluorescence proteins. The nomenclature used here indicates proteins fused with a fluorescence protein by a superscripted c for mCer3 and a superscripted v for Venus. The superscripted letters precede the protein names for amino-terminal, and follow them for carboxyl-terminal fusions. $^V$BH3-proteins were expressed by transient transfection of MCF-7 cells stably over-expressing $^C$Bcl-XL or $^C$Bcl-2 (*Figure 1*). Transient expression of $^V$BH3-proteins ensured that there was a large enough range of expression levels within the population of cells to collect the data needed to generate binding curves (*Aranovich et al., 2012*; *Kale et al., 2012*; *Liu et al., 2012*). The expressed fluorescent fusion proteins retained their expected anti- or pro-apoptotic properties (see below). To generate quantitative binding curves, TCSPC data were collected for $^C$Bcl-XL (FRET donor) and $^V$BH3-proteins (FRET acceptor), within regions of interest (ROIs) automatically identified in cells based on mCer3 signal intensity (*Figure 1A*). For each ROI the TCSPC data were used to determine the average fluorescence lifetime of the mCer3 donor and the fluorescence intensities of both the donor (mCer3) and the acceptor (Venus). Data from more than 8,500 ROIs were binned according to the ratio of Venus to mCer3 intensities and together with the corresponding lifetime data used to generate binding curves (*Figure 1B*).

### The impact of BH3 sequence mutations and small-molecule inhibitors on the interactions between anti-apoptotic proteins and BH3 only pro-apoptotic proteins

BH3-proteins engage the hydrophobic pocket of anti-apoptotic proteins through four conserved hydrophobic residues (h1-h4) within the BH3 region (*Chen et al., 2005*). ABT-263 prevents BH3-protein binding by competing for the binding of residues h2 and h4 to Bcl-XL (*Figure 1C*) (*Tse et al., 2008*). To quantify the importance of these residues in live cells, we expressed wild-type and mutant $^V$BH3-proteins with alanine mutations in the h2 and h4 positions (BH3-2A) and measured binding to $^C$Bcl-XL. As expected, BH3-2A mutations in $^V$Bad and $^V$tBid reduced binding to $^C$Bcl-XL dramatically (*Figure 1D–E*, compare black and red). However the BH3-2A mutant of $^V$BimEL bound to $^C$Bcl-XL similarly to wild-type $^V$BimEL (*Figure 1F*, compare black and red), suggesting interactions with other residues contribute to binding Bim to Bcl-XL. Consistent with this result ABT-263 displaced both $^V$Bad and $^V$tBid, but not $^V$BimEL from $^C$Bcl-XL (*Figure 1D–F*, compare green and cyan).

To quantify the impact of ABT-263 on $^V$BH3-proteins binding to $^C$Bcl-XL and $^C$Bcl-2, and maximize the dynamic range of the assay we estimated the FLIM-FRET efficiency from the fitted binding curves at intensity ratios of Venus to mCer3 of 0.5 and 0.25, respectively. The FRET signal from the non-binding mutant $^V$Bad4E (in which h1, h2, h3 and h4 in the BH3 region were mutated to glutamic acid) served as a negative control for FRET due to random collisions rather than binding, and was subtracted as background (*Figure 2A*). The percentage of the FRET signal remaining in the presence of ABT-263 is defined here as **R**esistance to displacement by ABT-263 (**R**$_{ABT-263}$, *Figure 2A*). The effects of mutations on the binding of BH3-proteins can be calculated similarly and for the BH3-2A mutation is reported as **R**$_{BH3-2A}$. R$_{ABT-263}$ and R$_{BH3-2A}$ values for all of the mutants analyzed are provided in *Table 1*.

Plotting R$_{ABT-263}$ for different interactions indicated that >80% of $^C$Bcl-XL:$^V$BimEL and $^C$Bcl-2:$^V$BimEL complexes were resistant to the addition of ABT-263 (*Figure 2B*). In contrast, $^C$Bcl-XL:$^V$Bad and $^C$Bcl-XL:$^V$tBid complexes were less than 50% resistant to the drug (*Figure 2B*-left, black) while the corresponding Bcl-2 complexes were less than 40% resistant (*Figure 2B*-left, blue, *Figure 2—figure supplement 1*). Calculated similarly, R$_{BH3-2A}$ values reflect the minimal effect of BH3-2A mutations on complexes of $^C$Bcl-XL and $^C$Bcl-2 with $^V$BimEL (R$_{BH3-2A}$ ~ 80%) compared to $^V$Bad (R$_{BH3-2A}$ ~ 2%) and $^V$tBid (R$_{BH3-2A}$ ~30%) complexes (*Figure 1D–F*, *Figure 2B*, and *Figure 2—figure supplement 1*).

To test the generality of these results in a different cell line that is resistant to the induction of cell death by BH3 proteins, complex formation was also assessed in Baby Mouse Kidney (BMK) cells in which the genes for expression of Bax and Bak were knocked out (DKO). These cells do not

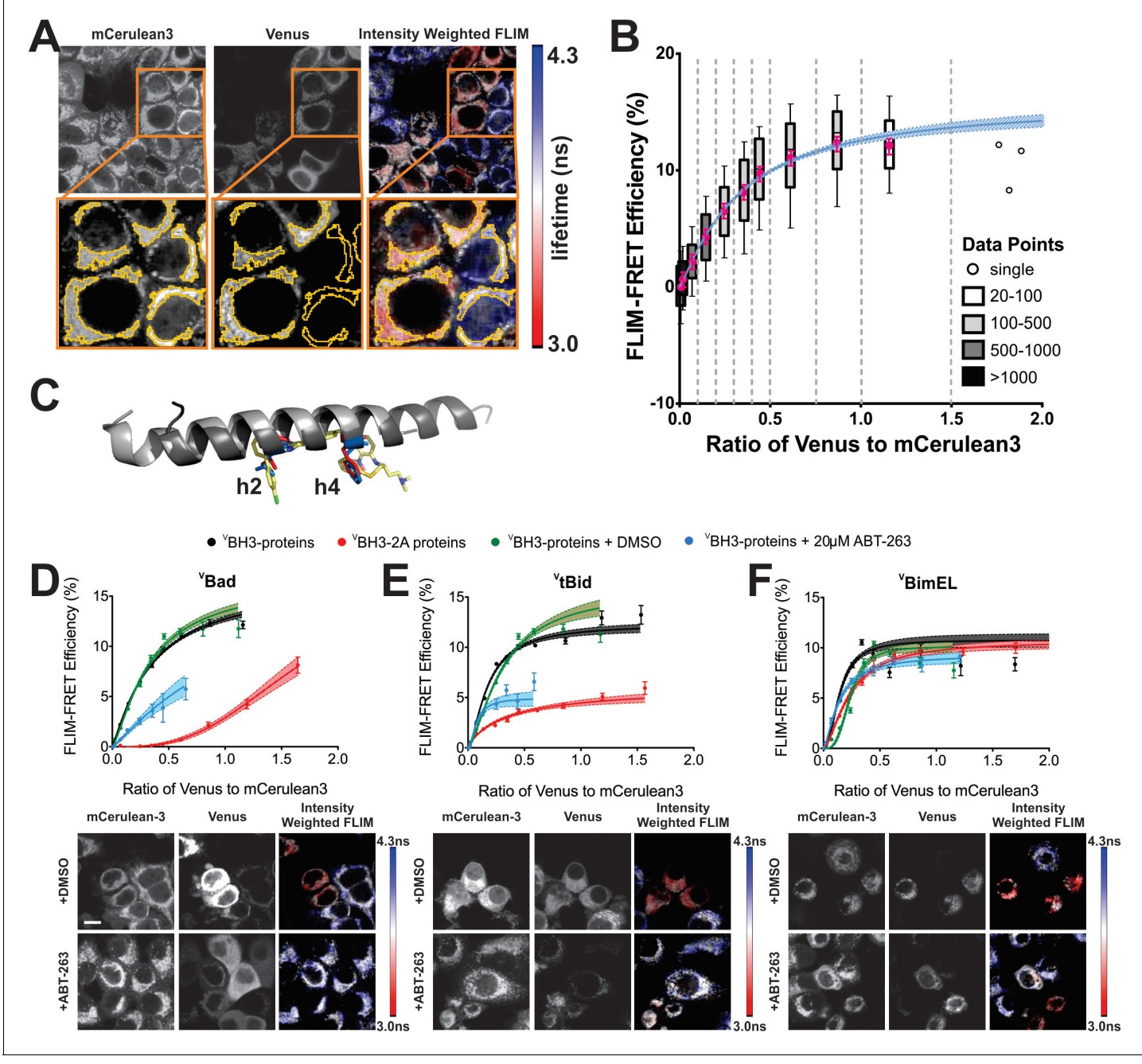

**Figure 1.** ABT-263 does not displace Bim from binding to Bcl-XL. (**A**) Live cell images of mCer3 and Venus intensities and intensity weighted FLIM images from FLIM-FRET measurements for the interaction between $^{C}$Bcl-XL and $^{V}$Bad. Higher magnification views are shown below with automatically identified Regions Of Interest (ROIs) outlined in yellow. Fluorescence lifetime images are presented in a continuous pseudo-color scale ranging from 3.0 to 4.3 ns. Intensity weighted fluorescence lifetime images were generated using ImageJ, both the contrast and gamma of the intensity image were adjusted so that the mitochondrial area can be easily visualized. Intensity weighted images are used only as an interpretive guide and were not used for any of the calculations reported. (**B**) Generation of binding curves. Data from more than 8,500 ROIs were binned according to the ratio of Venus to mCer3 intensities; mean (magenta dot) FLIM-FRET efficiency values for each bin were plotted versus the ratio of Venus to mCer3 intensity. The box encloses 50% of the data (25-75th percentiles), lines extend to the 10th and 90th percentiles and the number of ROIs in each bin (minimum 20) is indicated according to the scale at the right. The first bin is at zero and subsequent bins are separated by dotted lines. The means were fitted to a binding curve with a Hill slope for the interaction between $^{C}$Bcl-XL and $^{V}$Bad (blue line) with the 95% confidence interval for the optimal fit of the model to the data using GraphPad Prism shown (blue shaded area). (**C**) Structural alignment of ABT-263 with BH3 peptides (grey with h2 and h4 residues colored) from the complexes: Bcl-XL:Bim-BH3 (1PQ1, blue), Bcl-XL:Bad-BH3 (2BZW, red), and Bcl-XL:ABT-263 (4QNQ, yellow). ABT-263 and side chains of the key hydrophobic residues (**h2 and h4**) in the BH3 peptides are shown. (**D–F**) ABT-263 or mutation of h2 and h4 displaced $^{V}$Bad, $^{V}$tBid but not

*Figure 1 continued on next page*

*Figure 1 continued*

$^V$BimEL from $^C$Bcl-XL in MCF-7 cells. Binding curves for $^C$Bcl-XL and $^V$BH3-proteins: (**D**) $^V$Bad, (**E**) $^V$tBid, (**F**) $^V$BimEL. The samples were untreated (black), DMSO solvent control (green), 20 µM ABT-263 (cyan), or BH3-2A mutation in the BH3-protein (red). Representative images are shown below as labeled. Scale bar 10 µm. Curves were generated by fitting the data to a Hill equation; line width shading indicates 95% confidence interval for the fit. Individual points are the average FLIM-FRET efficiencies in corresponding bins (n ranges from 20 to 3000 in each bin combined from three independent experiments as in (**B**). Cyan lines are truncated because at higher ratios of Venus to mCer3 there is sufficient free BH3-protein to kill the cells. In all figures FLIM-FRET binding curves as shown in *Figure 1D–F* and figure supplements were fitted from data pooled from three independent experiments. Individual points in FLIM-FRET binding curves indicate the average FLIM-FRET efficiencies in corresponding bins (the number of data points ranges from 20 to 3000 in each bin), the error bars indicate standard error of the mean, and the dotted shadowed area for each curve represents the 95% confidence interval for the fit of the binding curve with a Hill slope to the data as in (**b**).Truncation of some of the curves is due to a lack of sufficient data at high expression levels of the acceptor proteins. This can be due to induction of apoptosis by the expressed protein or to limited expression of the exogenous protein in the transient transfections.

DOI: https://doi.org/10.7554/eLife.37689.003

The following source data is available for figure 1:

**Source data 1.** Source data fitted to Hill equations demonstrating that ABT-263 displaced tBid and Bad but does not displace Bim from binding to Bcl-XL.
DOI: https://doi.org/10.7554/eLife.37689.004

---

undergo apoptosis in response to expression of BH3-proteins and therefore retain normal cellular morphology. As expected, the $R_{ABT-263}$ and $R_{BH3-2A}$ values obtained in BMK-DKO cells for $^C$Bcl-XL and $^V$BH3-protein complexes were similar to those observed for MCF-7 cells ($^C$Bcl-XL (Black) in *Figure 2B* compared with *Figure 2C*, *Figure 1D–F* and *Figure 2—figure supplement 2*). Similar to the results in MCF-7 cells, the resulting $R_{ABT-263}$ and $R_{BH3-2A}$ values measured for $^C$Bcl-XL:$^V$Bad and $^C$Bcl-XL:$^V$tBid complexes ranged from 40% to 70% lower than those of $^C$Bcl-XL:$^V$BimEL complexes confirming that the unusual stability of $^C$Bcl-XL:$^V$Bim complexes is neither cell line-specific nor a result of molecular crowding that may occur when cells undergo apoptosis (*Figure 2c*). As an additional control to address this latter point directly lifetime measurements were compared for manually selected images of fully adherent MCF-7 cells expressing $^C$Bcl-XL and $^V$Bim-Bad that retained relatively normal morphology (alive cells) from the bottom of the imaging well. Images of cells with a more condensed morphology (dying cells) were subsequently recorded from a higher imaging plane. The distributions of lifetimes measured for the both sets of cells were very similar confirming that the changes in dying cells do not affect the lifetime measurements (*Figure 2—figure supplement 3*).

Compared to ABT-263 (IC50 >10 µM), the Bcl-XL specific inhibitors A-1155463 and A-1331852 more potently disrupted $^C$Bcl-XL:$^V$Bad complexes with R values of ~45% and 20% respectively, corresponding to an IC50 <1.25 µM for these compounds (*Figure 2D* and *Figure 2—figure supplement 4*). Nevertheless, their impact on $^C$Bcl-XL:$^V$BimEL complexes was limited, confirming the high stability of $^C$Bcl-XL:$^V$BimEL complexes. Results for displacement of BH3 proteins from Bcl-2 using the specific inhibitor venetoclax were uninterpretable due to interference with the FLIM measurements that resulted from the broad spectrum fluorescence of the drug. As a dual inhibitor, ABT-263 enables comparison of the effects of mutations for both $^C$Bcl-XL:$^V$BimEL and $^C$Bcl-2:$^V$BimEL complexes and was therefore used for subsequent experiments.

To confirm that the stability of the complexes measured in live cells was due to direct interactions, FRET was measured for donor-fluorophore-labeled recombinant BimL or tBid with acceptor-fluorophore-labeled Bcl-XL in a cell free liposome based system. In this assay BimL was used instead of BimEL due to the difficulty in purifying BimEL and the BH3-proteins were added at amounts predetermined to result in complete binding of BimL or tBid to Bcl-XL. ABT-263 was then titrated into the samples to compete with the BH3-proteins. As expected, a concentration of ABT-263 ~10 fold higher than that of Bcl-XL displaced tBid from Bcl-XL (IC50 ~400 nM) yet had little effect on the interaction between Bcl-XL and BimL (stable at >3 µM) (*Figure 2E*). Thus similar to what was seen in cells by FLIM FRET, direct binding of Bim to Bcl-XL is more resistant to ABT-263 than is binding of tBid to Bcl-XL.

To test the functional consequences of this we measured protein binding and liposome permeabilization at the same time. Under conditions that support measurement of both, the IC50 for ABT-263 for FRET between cBid and Bim with Bcl-XL was 212 ± 43 and>700 nM, respectively while for liposome permeabilization the corresponding EC50's were 65 ± 9 nM and 150 ± 16 nM, n = 3

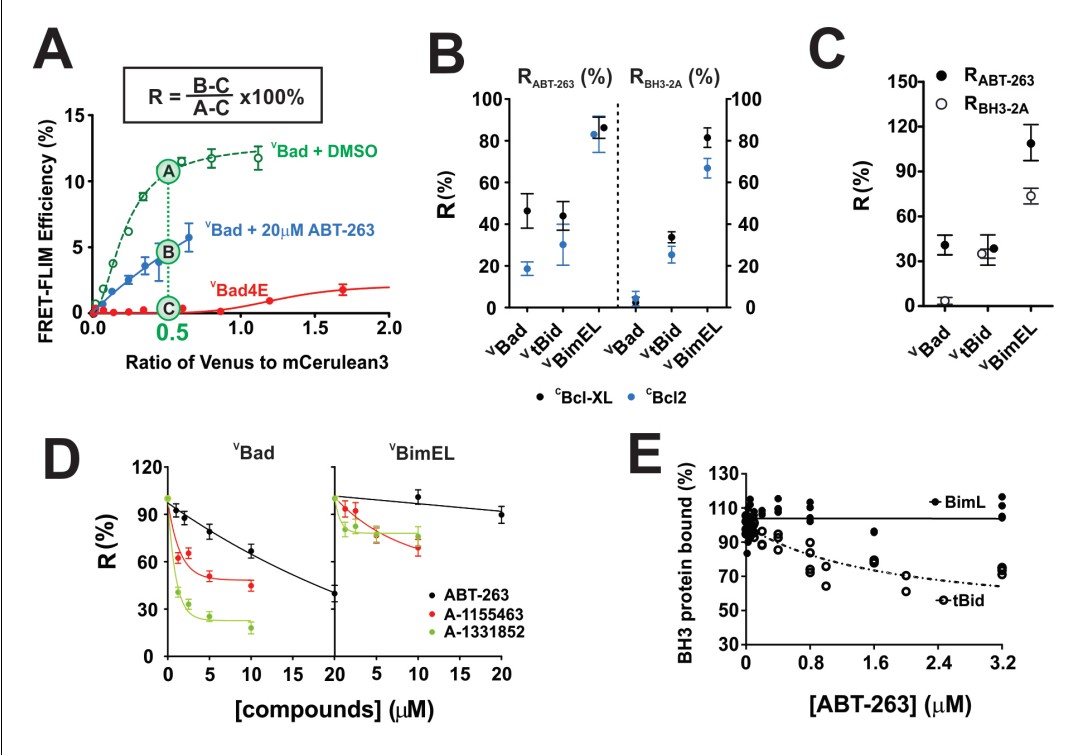

**Figure 2.** with four supplements: The impact of small-molecule inhibitors and BH3-sequence mutations on the interactions between anti-apoptotic proteins and BH3-only pro-apoptotic proteins. (**A**) Values of R can be used to define binding interactions from FLIM-FRET data. Sample calculation of $R_{ABT-263}$ for $^C$Bcl-XL:$^V$Bad binding in MCF-7 cells. To maximize the accuracy and dynamic range of the assay to quantify the impact of ABT-263 on the $^V$BH3-proteins binding to $^C$Bcl-XL, we interpolated the FLIM-FRET efficiency from the fitted binding curves at an intensity ratio of Venus to mCer3 of 0.5 (points A, (**B and C**). The non-binding mutant $^V$Bad4E, in which h1, h2, h3 and h4 in the BH3 region were all mutated to glutamic acid, served as a control for FRET due to random collisions (red line, point C) and was subtracted from A and B as background. The signal remaining after the addition of ABT-263 (cyan line, point B) expressed as a percentage of the signal with DMSO added instead of the drug (green line, point A) is defined as $R_{ABT-263}$. (**B**) $R_{ABT-263}$ and $R_{BH3-2A}$ for binding of the indicated $^V$BH3-proteins to $^C$Bcl-XL (black) from binding curves in shown in **Figure 1and** $^C$Bcl-2 (blue) in MCF-7 cells. $R_{BH3-2A}$ is calculated similarly to $R_{ABT-263}$ except the FLIM-FRET efficiency of the mutant is substituted for the value after adding ABT-263. Binned data, binding curves and sample images for Bcl-2 are shown in **Figure 2—figure supplement 1**. (**C**) The $R_{ABT-263}$ and $R_{BH3-2A}$ values for $^C$Bcl-XL:$^V$BH3 proteins interactions in BMK-DKO cells. Binned data, binding curves and sample images are shown in **Figure 2—figure supplement 2**. Control experiments showing that the morphology changes that accompany cell death do not change the lifetime values determined by FLIM are shown in **Figure 2—figure supplement 3**. (**D**) Bcl-XL inhibitors displace $^V$Bad efficiently but $^V$BimEL poorly from $^C$Bcl-XL in live cells. The dose-dependent inhibition curves due to the indicated concentrations of ABT-263 (black), and the Bcl-XL inhibitors A-1155463 (red) and A-1331852 (green) shown for $^C$Bcl-XL:$^V$Bad (left) and $^C$Bcl-XL:$^V$BimEL (right) complexes in live MCF-7 cells. R values are ±95% confidence intervals from binned data and binding curves shown in **Figure 2—figure supplement 4** (**E**) ABT-263 displaces tBid (10 nM) but not Bim (10 nM) from Bcl-XL (40 nM) in vitro. Percent of BH3 protein bound to Bcl-XL (BH3 bound %) measured by loss of FRET for Bcl-XL:tBid and Bcl-XL:Bim quantified for purified full-length dye labeled proteins incubated with liposomes and the indicated concentrations of drug. Data are from three experimental replicates, not all points are visible due to overlap. In all figures $R_{ABT-263}$ and $R_{BH3-2A}$ data points for Bcl-XL and Bcl-2 are black and blue respectively.

DOI: https://doi.org/10.7554/eLife.37689.005

The following source data and figure supplements are available for figure 2:

**Source data 1.** Source data for the impact of small-molecule inhibitors and BH3-sequence mutations on the interactions between anti-apoptotic proteins and BH3-only pro-apoptotic proteins.
DOI: https://doi.org/10.7554/eLife.37689.006

**Figure supplement 1.** FLIM-FRET binding curves for Bcl-2 binding to BH3 proteins in MCF-7 cells.
DOI: https://doi.org/10.7554/eLife.37689.007

**Figure supplement 1—source data 1.** Source data fitted to a Hill equation to generate FLIM-FRET binding curves for Bcl-2 binding to BH3 proteins in MCF-7 cells.
DOI: https://doi.org/10.7554/eLife.37689.008

**Figure supplement 2.** FLIM-FRET binding curves for Bcl-XL binding to BH3 proteins in BMK DKO cells.
DOI: https://doi.org/10.7554/eLife.37689.009

*Figure 2 continued on next page*

*Figure 2 continued*

**Figure supplement 2—source data 1.** Source data fitted to a Hill equation to generate FLIM-FRET binding curves for Bcl-XL binding to BH3 proteins in BMK DKO cells.

DOI: https://doi.org/10.7554/eLife.37689.010

**Figure supplement 3.** FLIM-FRET measurements for <sup>C</sup>Bcl-XL binding to <sup>V</sup>BimEL-Bad in alive and dying MCF-7 cells.

DOI: https://doi.org/10.7554/eLife.37689.011

**Figure supplement 4.** FLIM-FRET binding curves for <sup>V</sup>Bad (left) and <sup>V</sup>BimEL (right) to <sup>C</sup>Bcl-XL in MCF-7 cells in the presence of different concentrations of BH3-mimetics.

DOI: https://doi.org/10.7554/eLife.37689.012

**Figure supplement 4—source data 1.** Source data fitted to a Hill equation to generate FLIM-FRET binding curves for Bad and BimEL to Bcl-XL in MCF-7 cells.

DOI: https://doi.org/10.7554/eLife.37689.013

---

independent duplicates,±standard error. Thus, the change in binding affinity results in a corresponding change in membrane permeabilization.

## Regulation of apoptosis by fluorescent fusion proteins in live cells

The functional properties of the fusion proteins used in FLIM-FRET experiments were examined using multiple assays in MCF-7 cells. Transient transfection by the requisite plasmids demonstrated that expression of $^V$Bim and $^V$tBid killed MCF-7 cells as assessed morphologically for both cell and nuclear condensation and mitochondrial transmembrane potential (*Figure 3A*). In contrast, expression of $^V$Bad had little effect on cell death in this cell line suggesting that inhibition of Bcl-2 and Bcl-XL is not sufficient to induce apoptosis in MCF-7 cells.

To examine the effect of expression of exogenous $^C$Bcl-XL (confirmed by immunoblotting in *Figure 3B*) apoptosis was triggered with TNFα and cycloheximide (CHX) in the same cell clones used for FLIM-FRET above. Untreated and cells treated with only CHX were used as negative controls. As expected, untransfected MCF-7 cells were efficiently killed while cells expressing either $^C$Bcl-XL or $^C$Bcl-2 were highly resistant to this well-established method for the induction of apoptosis (*Figure 3C*).

To determine the importance of other anti-apoptotic proteins expressed endogenously in these cells, they were treated with specific small molecule inhibitors. Consistent with the results demonstrating resistance to Bad, MCF-7 cells were largely resistant to inhibition of Bcl-2 and/or Bcl-XL (*Figure 3D*). Neither selective nor dual inhibition of Bcl-2 and Bcl-XL induced substantial cell death in MCF-7 cell lines (all below 20% Dead) demonstrating these cells are not highly dependent on expression of either of these anti-apoptotic proteins. Nevertheless, expression of $^C$Bcl-2 or $^C$Bcl-XL reduced cell death further (*Figure 3D*).

In contrast, MCF-7 cells were sensitive to inhibition of MCL-1 (*Figure 3E*). Moreover, overexpression of $^C$Bcl-XL or $^C$Bcl-2 protected MCF-7 cells from the MCL-1 inhibitor S-63845, likely by inhibiting pro-apoptotic proteins displaced by the BH3-mimetic from MCL-1 (*Figure 3E*). Taken together the data demonstrate that MCF-7 cells are primarily dependent on expression of MCL-1 for survival and that the exogenously expressed fluorescence protein fusions function as predicted in this cell line.

## The Bim BH3 and Bim CTS both contribute to resistance to ABT-263

It is commonly accepted that the only stable binding interaction between Bim and anti-apoptotic proteins is via the BH3-region of Bim (Bim BH3) binding in the hydrophobic groove formed by the BH3 1–2 regions of the anti-apoptotic proteins (*Liu et al., 2010*; *Sattler et al., 1997*). However, resistance to ABT-263 suggests that in the context of the whole protein either the Bim BH3 region binds differently than BH3-peptides to Bcl-XL or a region not included in previous in vitro experiments, such as the N-terminal and C-terminal sequences shared by all three isoforms, contains sequences conferring ABT-263 resistant binding. To test these sequences, mutants harboring deletions in $^V$BimEL and mutants in which the BH3-regions were exchanged between BimEL and Bad (*Figure 4A*) were assayed by FLIM-FRET for resistance to ABT-263 and to the BH3-2A mutation (*Figure 4B–C* and *Figure 4—figure supplement 1*).

The trends observed for binding of the mutants to $^C$Bcl-XL and $^C$Bcl-2 were remarkably similar. Truncation of the N-terminus of BimEL ($^V$BimEL-dN) had little impact on Bim binding (*Table 1*), but

**Table 1.** Values of $R_{BH3-2A}$ and $R_{ABT-263}$ for all reported interactions.

| Cell lines Constructs | [c]Bcl-XL $R_{BH3-2A}$ Value | 95% CI | $R_{ABT-263}$ Value | 95% CI | [c]Bcl-2 $R_{BH3-2A}$ Value | 95% CI | $R_{ABT-263}$ Value | 95% CI |
|---|---|---|---|---|---|---|---|---|
| [V]Bad | 2.44 | 2.50 | 46.36 | 8.23 | 4.33 | 3.48 | 18.63 | 3.24 |
| [V]tBid | 33.70 | 2.60 | 43.95 | 6.84 | 25.33 | 3.98 | 30.16 | 9.82 |
| [V]BimEL | 81.43 | 4.65 | 86.26 | 5.10 | 66.85 | 4.70 | 83.06 | 8.69 |
| [V]Bad-Bim | 70.19 | 4.03 | 97.96 | 6.45 | 54.91 | 3.92 | 65.46 | 7.40 |
| [V]BimEL-dN | 101.87 | 4.37 | 90.81 | 7.01 | 82.57 | 4.27 | 90.47 | 10.25 |
| [V]BimEL-Bad | 26.12 | 2.70 | 60.55 | 4.43 | 32.74 | 5.80 | 65.75 | 6.36 |
| [V]BimEL-dCTS | 15.08 | 3.17 | 69.05 | 4.57 | 13.60 | 2.21 | 60.02 | 6.73 |
| [V]BimEL-Bad-dCTS | 10.61 | 4.26 | 40.25 | 5.67 | 11.95 | 4.07 | 12.51 | 6.60 |
| [V]Bad-Bim-WAA | ND | ND | 78.51 | 5.81 | ND | ND | 50.15 | 7.22 |
| [V]Bad-Bim-I146Y | 46.56 | 3.20 | 83.24 | 5.34 | 26.78 | 4.21 | 53.17 | 5.43 |
| [V]Bad-Bim-Q148R | 74.45 | 4.24 | 89.82 | 6.21 | 56.94 | 5.46 | 53.46 | 7.33 |
| [V]Bad-Bim-I153M | 96.53 | 4.60 | 92.78 | 6.00 | 102.30 | 8.94 | 68.51 | 7.51 |
| [V]Bad-Bim-I146Y-Q148R | 50.46 | 3.46 | 90.18 | 7.59 | 37.73 | 3.88 | 61.77 | 7.54 |
| [V]Bad-Bim-I146Y-I153M | 56.80 | 3.27 | 88.97 | 6.12 | 42.59 | 4.33 | 69.07 | 7.73 |
| [V]BimEL-I146Y | 70.99 | 7.58 | 88.66 | 10.85 | 54.17 | 3.49 | 65.32 | 8.72 |
| [V]BimEL-I146Y-dCTS | 12.29 | 2.89 | 38.45 | 8.31 | 8.57 | 4.03 | 52.86 | 6.02 |
| [V]BimEL-L185E | 26.68 | 3.62 | 41.85 | 10.24 | 16.09 | 3.87 | 51.02 | 12.29 |
| [V]BimEL-I146Y-L185E | 19.44 | 4.10 | 40.01 | 8.07 | 5.79 | 9.81 | 36.85 | 8.81 |
| [V]BimEL-CTS2A | 77.99 | 4.43 | 88.79 | 6.24 | 60.24 | 3.77 | 80.69 | 11.40 |
| [V]BimEL-I181E | 49.19 | 50.23 | 63.86 | 6.16 | 47.20 | 2.96 | 73.26 | 10.45 |
| [V]BimEL-L185E | 26.68 | 3.62 | 41.85 | 10.24 | 16.09 | 3.87 | 51.02 | 12.29 |
| [V]BimEL-I188E | 67.17 | 5.14 | 50.96 | 7.66 | 47.47 | 3.36 | 63.04 | 8.04 |
| [V]BimEL-V192E | 52.57 | 53.59 | 70.14 | 7.64 | 54.50 | 3.56 | 64.84 | 9.70 |
| [V]BimEL-L185A | 159.10 | 14.88 | | | | | | |
| [V]BimEL-L185D | 20.31 | 5.29 | | | | | | |
| [V]BimEL-L185E | 26.16 | 3.57 | | | | | | |
| [V]BimEL-L185F | 99.99 | 5.15 | | | | | | |
| [V]BimEL-L185G | 86.82 | 7.91 | | | | | | |
| [V]BimEL-L185H | 49.10 | 4.50 | | | | | | |
| [V]BimEL-L185M | 89.98 | 4.42 | | | | | | |
| [V]BimEL-L185P | 58.95 | 3.61 | | | | | | |
| [V]BimEL-L185R | 48.08 | 3.33 | | | | | | |
| [V]BimEL-L185S | 73.49 | 4.67 | | | | | | |
| [V]BimEL-L185stop | 26.87 | 4.94 | | | | | | |
| BimEL[V] | | | 89.41 | 5.80 | | | | |

DOI: https://doi.org/10.7554/eLife.37689.014

replacing the BH3 region in Bad with that in BimEL ([V]Bad-BimEL) increased $R_{ABT-263}$ compared with [V]Bad. Consistent with this result, binding of the reciprocal mutant ([V]BimEL-Bad) to either [C]Bcl-XL or [C]Bcl-2 was more sensitive to the drug and the BH3-2A mutation ($R_{ABT-263}$ ~60–65%, **Figure 4B** $R_{BH3-2A}$ ~30% **Figure 4C**, black and blue) respectively, compared to [V]BimEL (both R values ~ 80%, **Table 1**). As expected, deletion of the entire BH3 region (BimEL-d20, **Figure 4A**) reduced the interactions with Bcl-XL and Bcl-2 to a level similar to random collisions in the membrane ($R_{d20}$ ~25%, **Figure 4—figure supplement 1F and L**). Unexpectedly, deletion of the C-terminal sequence (CTS)

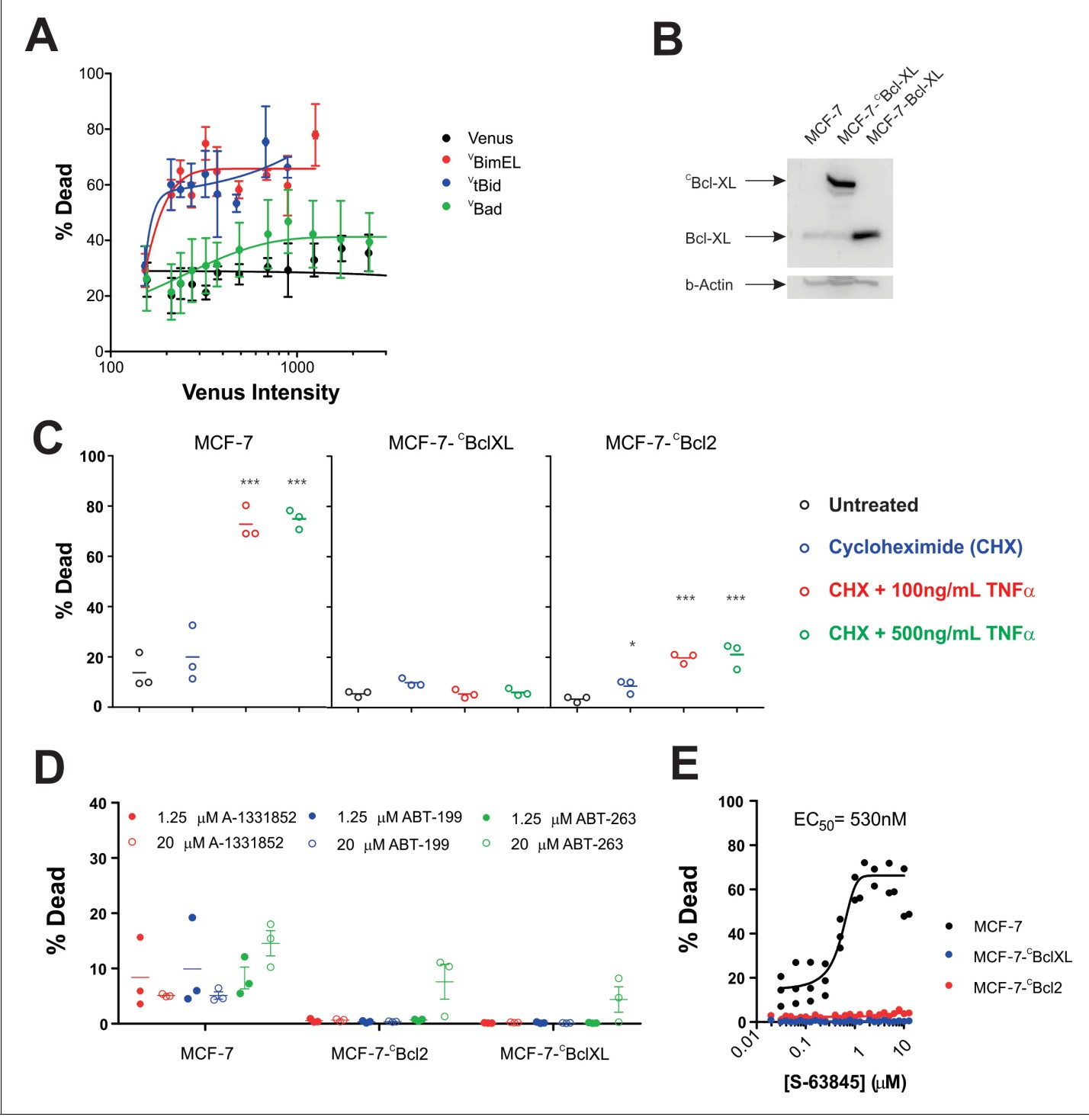

**Figure 3.** MCF-7 cells are killed by transient expression of BimEL or Bid, are protected by stably expressed Bcl-XL or Bcl-2 and depend on MCL-1 for survival. (**A**) Transient expression of exogenous $^V$tBid (blue) and $^V$BimEL (red) killed MCF-7 cells. MCF-7 cells were largely resistant to expression of $^V$Bad (green) as the resulting cell death was similar to expression of the control protein Venus (black). Images of individual cells were assessed for apoptosis based on staining with TMRE and the nuclear dye DRAQ five using a multiparametric linear classifier. An increase in the percentage of cells scored as dead or dying (% Dead) as a function of Venus intensity demonstrated that the $^V$tBid and $^V$BimEL fusion proteins retain pro-apoptotic activity. Error bars indicate standard error for three independent replicates. At least 30 cells were analyzed at each point representing a Venus intensity bin. (**B**) Immunoblotting of lysates from MCF-7 cells (lane 1) and MCF-7 cells expressing exogenous $^C$Bcl-XL (lane 2) or Bcl-XL (Lane 3) with an antibody to Bcl-XL demonstrated that the exogenous proteins are at least 20-fold over-expressed compared to endogenous Bcl-XL. The same blot was probed for β-actin as a loading control. (**C**) Cells classified as dead or dying (% Dead) for 3 cell lines (MCF-7 and MCF-7 expressing either $^C$Bcl-XL or $^C$Bcl-2), treated

*Figure 3 continued on next page*

*Figure 3 continued*

with 2 µg/ml cyclohexamide (CHX), or CHX plus TNFα (250 ng/ml and 500 ng/ml) for 24 hr. Data (% Dead) for three independent replicates (circles) and the mean of the replicates (line) are plotted. A one-way ANOVA test was performed with a Dunnett's Multiple Comparison post-test (Graphpad Prism), to compare all treated wells with untreated controls for each cell line. (D) Overexpression of $^C$Bcl-2 and $^C$Bcl-XL protected cells from BH3-mimetics. Points represent the average percentage of cells classified as dead or dying (% Dead) for individual replicates, with the mean of the replicates indicated by a line. MCF-7 cells and MCF-7 cells expressing either Bcl-2 or Bcl-XL were treated with 1.25 µM (dot) and 20 µM (circle) ABT-199, A-1331852 or ABT-263 as indicated above. Neither selective nor dual inhibition of Bcl-2 and Bcl-XL induced substantial cell death in MCF-7 cell lines (all below 20% Dead) demonstrating these cells are not highly dependent on expression of either of these anti-apoptotic proteins. Nevertheless expression of Bcl-2 or Bcl-XL reduced cell death to barely detectable levels. (E) The MCL-1 inhibitor, S-63845 (Servier) kills MCF-7 (black), but does not kill MCF-7 $^C$Bcl-XL (blue) and MCF-7 $^C$Bcl-2 (red) cells. An EC50 value of 530 ± 6 nM, was calculated for MCF-7 cells in GraphPad Prism using a non-linear fit of normalized data on a log scale (log(agonist) verses response, and variable slope (four parameters)).

DOI: https://doi.org/10.7554/eLife.37689.015

The following source data is available for figure 3:

**Source data 1.** Multiparametric source data for MCF-7 cell death in response to transient expression of BimEL or Bid and the protection afforded by stably expressed Bcl-XL or Bcl-2 and the dependence of MCF-7 cells on MCL-1 for survival.

DOI: https://doi.org/10.7554/eLife.37689.016

resulted in significant decreases in $R_{ABT-263}$ and $R_{BH3-2A}$ (*Figure 4B–C* and *Table 1*). Combining deletion of the Bim CTS with replacement of the Bim BH3 region by that of Bad ($^V$BimEL-Bad-dCTS), reduced the values of $R_{ABT-263}$ and $R_{BH3-2A}$ to that of $^V$Bad (*Figure 4B–C* and *Table 1*). Thus, both BH3 and CTS sequences contribute to Bim binding anti-apoptotic proteins.

## The CTS of Bim contributes to pro-apoptotic activity

To examine the functional importance of the Bim CTS further, the pro-apoptotic activities of selected Bim mutants were examined in MCF-7 cells with and without exogenously-expressed Bcl-XL or Bcl-2. Expression of $^C$Bcl-XL or $^C$Bcl-2 efficiently protected cells from lower levels of expression of $^V$BimEL and $^V$BimEL-dCTS (<1000 units of Venus intensity) however, in neither case was the protection complete at higher levels of expression (*Figure 5A*).

Unexpectedly, in MCF-7 cells there was no significant difference in induction of cell death by removing the CTS from BimEL (*Figure 5A*). There are two factors that are likely contributing to this result. First. Bim not only inhibits anti-apoptotic proteins it also activates the executioner proteins Bax and Bak, potentially confounding analysis. As this mechanism may be relevant in multiple cell lines it is examined in more detail elsewhere (*Chi et al., 2019*). Second, MCF-7 cells depend on MCL-1 for survival (*Figure 3E*).

To avoid the contribution to cell death by inhibition of MCL-1 we made use of the Bim mutant with the Bad BH3 region ($^V$BimEL-Bad) as it is not expected to inhibit Mcl-1. Consistent with expectation, $^V$BimEL-Bad was less effective than $^V$BimEL in killing MCF-7 cells (*Figure 5B*). Deletion of the CTS from $^V$BimEL-Bad resulted in a further loss of pro-apoptotic activity suggesting that in $^V$BimEL the effect of the CTS was masked by $^V$BimEL binding to and inhibiting MCL-1 (*Figures 5A* and *3b*). Consistent with the results for $^V$BimEL, inhibition of $^V$BimEL-Bad by Bcl-2 and Bcl-XL was limited. While this result is also consistent with a role for MCL-1 in binding pro-apoptotic proteins displaced from Bcl-2 and Bcl-XL by $^V$BimEL-Bad there are other possible explanations that will be the subject of future investigations.

To examine potential molecular mechanisms underlying the effects of deletion of the CTS from Bim in a more controlled system we examined BimL and BimL-dCTS binding to Bcl-XL and displacement of BH3 proteins from Bcl-XL using purified proteins and the FRET based assays similar to those described above (*Figure 2E*). When assayed directly, in incubations containing mitochondria, deletion of the CTS from BimL reduced the apparent affinity for binding to Bcl-XL (*Figure 5C*). As predicted from the reduced affinity of this interaction, BimL-dCTS was also less effective at displacing tBid and activated Bax from Bcl-XL (*Figure 5D and E*, respectively). Control FRET measurements indicated that at the concentrations used binding of cBid by Bcl-XL was saturated and that all of the proteins other than BimL-dCTS were bound to liposome membranes. Moreover, the interactions take place on the membrane surface as both tBid and activated Bax very efficiently bind to liposomes (*Kale et al., 2014*).

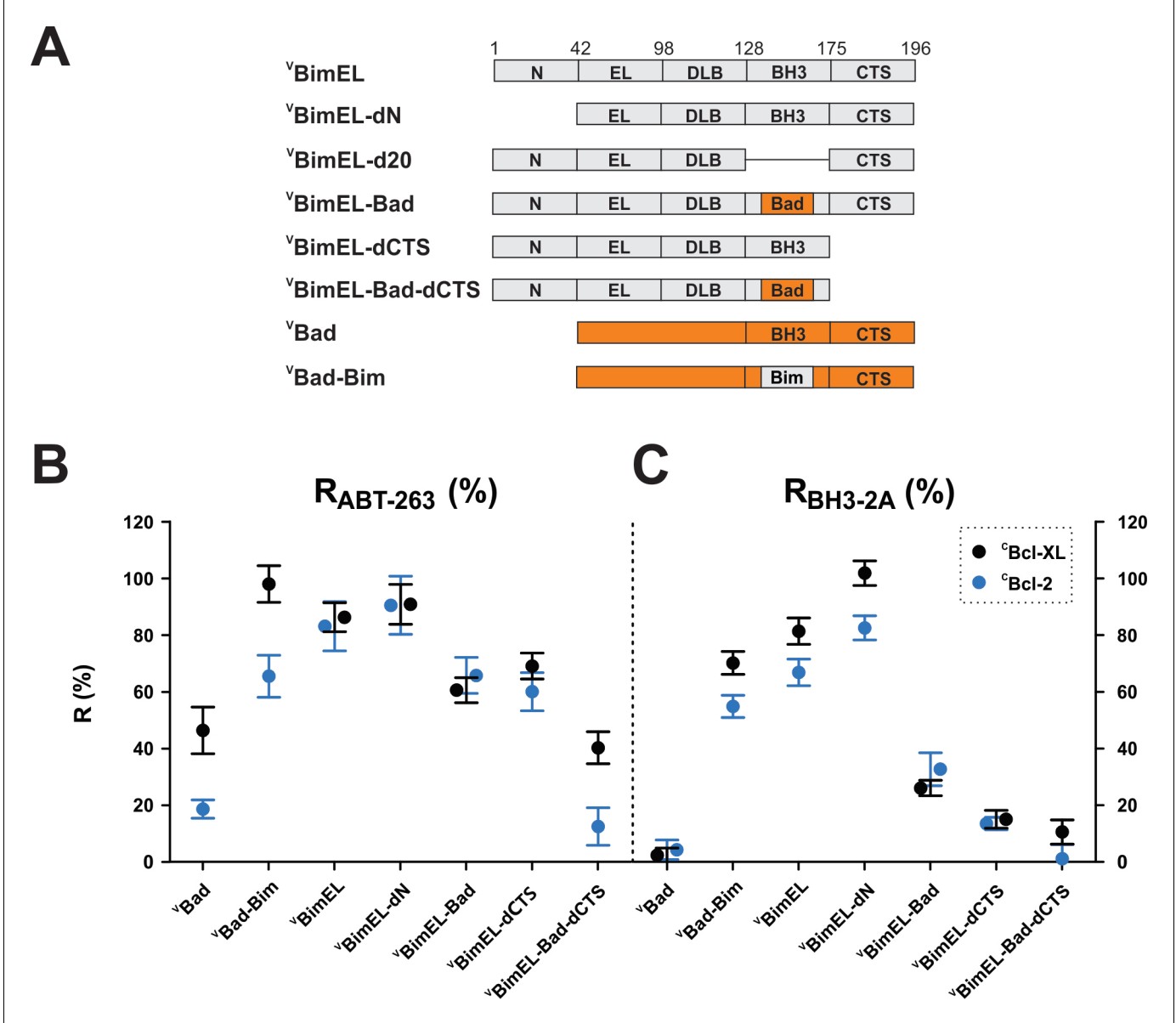

**Figure 4.** Resistance of $^C$Bcl-XL:$^V$BimEL and $^C$Bcl-2:$^V$BimEL complexes to ABT-263 is dependent on both the Bim BH3 and the Bim CTS. (**A**) Stick diagrams for the different $^V$BimEL constructs expressed in cells. (**B**) $R_{ABT-263}$ for $^C$Bcl-XL:$^V$BH3-protein complexes (black) and $^C$Bcl-2:$^V$BH3-protein complexes (blue). (**C**) $R_{BH3-2A}$ of $^C$Bcl-XL:$^V$BH3-protein complexes (black) and $^C$Bcl-2:$^V$BH3-protein complexes (blue). Data in (**B**) and (**C**) are mean ±95% confidence intervals calculated as in *Figure 2* from FLIM-FRET binding curves shown in *Figure 4—figure supplement 1*. Data for $^V$Bad and $^V$BimEL from *Figure 2* are included to facilitate direct visual comparisons. All binding data are from *Table 1*.

DOI: https://doi.org/10.7554/eLife.37689.017

The following source data and figure supplements are available for figure 4:

**Source data 1.** Source data for the calculation of R values for resistance of Bcl-XL:BimEL and Bcl-2:BimEL complexes to ABT-263 for the various mutant BH3 proteins.
DOI: https://doi.org/10.7554/eLife.37689.018
**Figure supplement 1.** Mutations in the BH3 region and the Bim CTS impair Bim binding to Bcl-XL and Bcl-2.
DOI: https://doi.org/10.7554/eLife.37689.019
**Figure supplement 1—source data 1.** Source data fit to a Hill equation to determine how mutations in the BH3 region and the Bim CTS impair Bim binding to Bcl-XL and Bcl-2.
DOI: https://doi.org/10.7554/eLife.37689.020

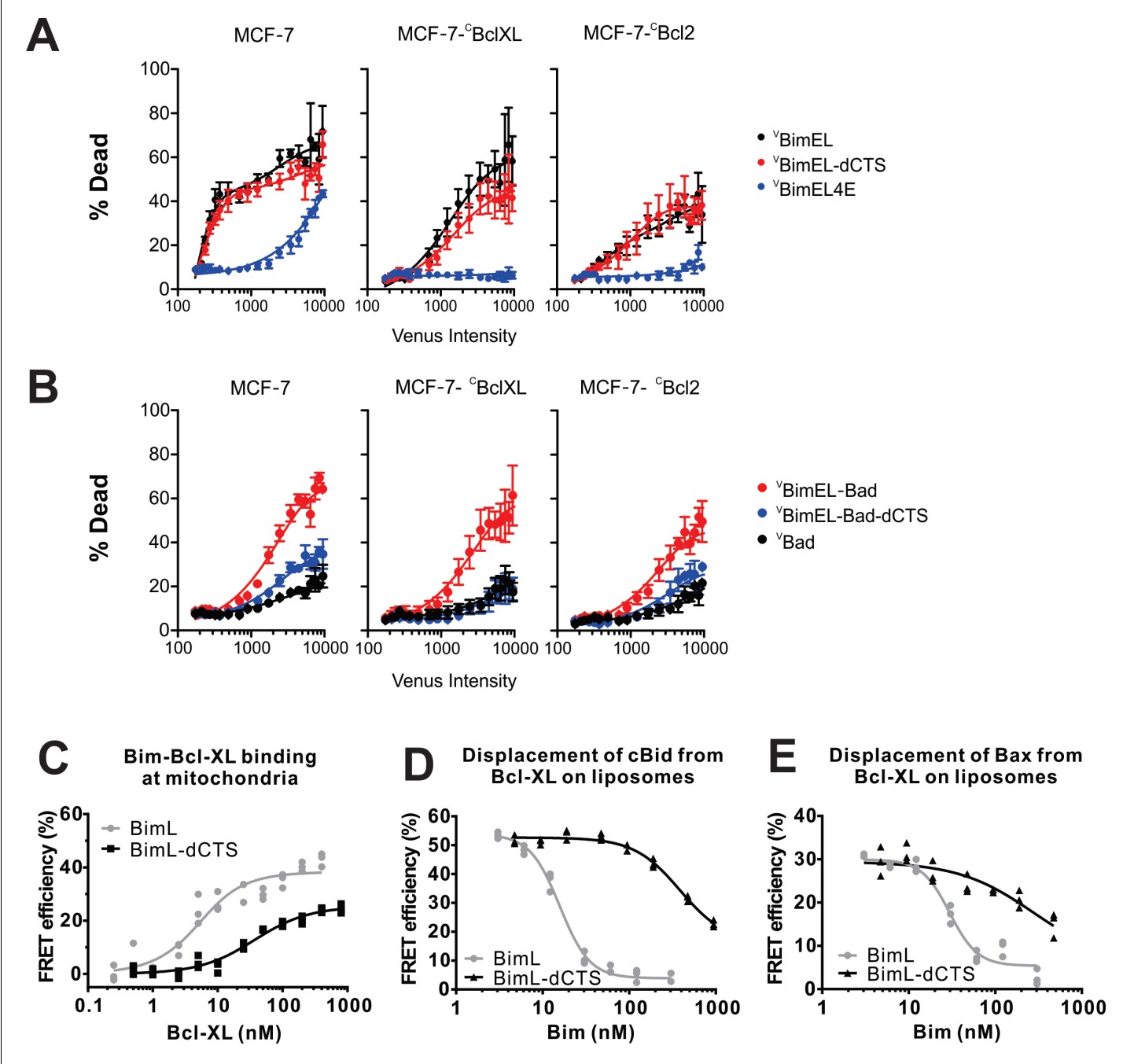

**Figure 5.** The CTS contributes to Bim mediated inhibition of Bcl-XL and Bcl-2. The BH3-proteins indicated to the right of the panels were expressed by transient transfection in MCF-7 cells and MCF-7 cells stably expressing either CBcl-XL or CBcl-2, as indicated. Commitment to cell death (% Dead) was assessed for individual cells using a linear classifier trained on both cell and nuclear morphology and TMRE intensity. Venus intensity was used to estimate for individual cells the relative BH3 protein expression and to divide them into bins with similar expression levels. For each bin, the % of cells classified as dead or dying (% Dead) is represented by the means from three replicates (colored dots); bars indicate standard error of the means. (A) Transient expression of pro-apoptotic BH3 proteins killed MCF-7 cells in an expression level dependent manner that was inhibited by exogenous expression of CBcl-XL and CBcl-2 (black lines). Deletion of the CTS of BimEL did not inhibit induction of cell death resulting from expression of BimEL in all 3 cell lines (red lines). High level expression of BimEL-4E killed MCF-7 cells in a manner completely inhibited by exogenous expression of either anti-apoptotic protein (navy blue lines). (B) The BH3 protein VBimEL-Bad that is unable to bind MCL-1 has reduced cell killing activity in MCF-7 cells that depends on the BIM CTS sequence and is poorly inhibited by exogenous expression of Bcl-XL or Bcl-2. Expression level dependent cell death was equivalent for VBimEL-Bad-dCTS and Bad consistent with MCF-7 cells being protected by endogenous MCL-1. (C–E) Interactions of Bcl-XL with BimL, BimL-dCTS, tBid and Bax measured using FRET. Data from three independent experiments are shown as individual points, some of which are not

*Figure 5 continued on next page*

Figure 5 continued

visible due to overlap. (C) The CTS of BimL increases its affinity for Bcl-XL. Binding of 4 nM Alexa568-labeled BimL or BimL-dCTS to the indicated amounts of Alexa647-labeled Bcl-XL was measured by FRET in samples containing mouse liver mitochondria. The resultant apparent Kd values for binding to Bcl-XL by BimL and BimL-dCTS, 3 ± 1 nM and 35 ± 5 nM, respectively are both well below the concentrations of the proteins expressed in cells. (D) The CTS of BimL increases displacement of Bid from Bcl-XL. Binding of cBid to membranes displaces the N-terminal region and the remaining tBid portion containing the BH3 region binds to membranes and anti-apoptotic proteins. Complexes of Alexa568-labeled cBid (4 nM) and Alexa647-labeled Bcl-XL (10 nM) were formed in incubations containing 2.9 nM liposomes. Displacement of cBid from Bcl-XL was measured by loss of FRET between Alexa568-labeled cBid and Alexa647-labeled Bcl-XL upon addition of the indicated concentrations of BimL or BimL-dCTS. (E) The CTS of BimL increases displacement of Bax from Bcl-XL. Bim mediated displacement of Bax from Bcl-XL was measured as in (d) except that complexes between Bcl-XL and Bax were assembled from Alexa568-labeled Bax (10 nM), a Bid mutant that activates Bax but does not bind Bcl-XL (10 nM cBidmt1), Alexa647-labeled Bcl-XL (30 nM), 2.9 nM liposomes and the indicated concentrations of BimL or BimL-dCTS. Because in these reactions 30 nM Bcl-XL was required to bind 10 nM Bax saturably as opposed to the 10 nM Bcl-XL used in (d) the concentrations of BimL and BimL-dCTS needed to displace Bax are higher than for displacement of tBid in experiments with added cBid. The curves are further complicated because the displaced Bax can also bind to BimL and BimL-dCTS. Thus while these data demonstrate that both BimL and BimL-dCTS displace Bax from Bcl-XL the curves are difficult to interpret in terms of binding constants.

DOI: https://doi.org/10.7554/eLife.37689.021

The following source data is available for figure 5:

**Source data 1.** Multiparametric cell death data for the mutants demonstrating that the Bim CTS contributes to Bim mediated inhibition of Bcl-XL and Bcl-2.

DOI: https://doi.org/10.7554/eLife.37689.022

## Multiple residues in the Bim BH3 contribute to ABT-263 resistance

There are five hydrophobic residues/groups named h0-h4 on the hydrophobic side of the BH3 α-helix in pro-apoptotic BH3-proteins hypothesized to mediate BH3-protein binding to anti-apoptotic proteins (*Figure 6A*). To quantify the binding due to each of these residues without potential interference from the additional binding site within the Bim CTS, mutations were introduced into a chimeric fusion protein composed of the Bad sequence with the Bim BH3 region (VBad-Bim in *Figure 4A*). The introduced mutations substitute residues in the Bim BH3 sequence at positions h0, h1, h1 +2 (an R between h1 and h2), and h3 with the corresponding residues in Bad (*Figure 6A*). When measured by FLIM-FRET, the relative effects of most of the individual mutations on the $R_{ABT-263}$ and $R_{BH3-2A}$ values for VBad-Bim binding to CBcl-XL (black) and CBcl-2 (blue) reduced binding by 10–20% (*Figure 6B*, *Figure 6—figure supplement 1* and *Table 1*). Among these the h0 and h1 residues contributed to Bim binding to Bcl-2 and Bcl-XL the most. Substitution of the h0 sequence with WAA reduced $R_{ABT-263}$ for VBad-Bim to CBcl-XL from 98% (95% Confidence Interval (CI) 6%) to 79% (CI 6%). Similarly, substitution of I146 with Y decreased $R_{ABT-263}$ for VBad-Bim to 83% (CI 5%). The reductions in $R_{ABT-263}$ for VBad-Bim to Bcl-2 were similar (*Table 1*). However, the $R_{ABT-263}$ for VBad-Bim binding to both Bcl-XL and Bcl-2 was unexpectedly high. As Bad also has a CTS sequence that binds to membranes and may have other as yet uncharacterized functions we examined the effect of the I146Y mutation in BimEL-dCTS. The $R_{ABT-263}$ for Bcl-XL for this mutant (BimEL-I146Y-dCTS) was 38% (CI 8%) compared to 69% (CI 5%) for VBimEL-dCTS binding to CBcl-XL suggesting a role for I146 in BimEL binding to Bcl-XL (*Figure 6C*). Although a reduction was also seen for CBcl-2 the confidence intervals overlapped suggesting there are subtle differences in Bim binding to Bcl-XL and Bcl-2. The small differences observed for substitutions at other positions within Bim-BH3 suggest either we have reached the limit of sensitivity for the FLIM-FRET assay or they may augment but individually are not critical for ABT-263 resistant binding of Bim to Bcl-XL and Bcl-2.

## The Bim CTS increased Bim binding to Bcl-XL and Bcl-2 independent of Bim binding membranes

Deletion of the Bim CTS has been shown to abrogate binding to membranes (*O'Connor et al., 1998*). We therefore hypothesized that the membrane binding function of this region might affect the resistance of Bcl-XL/Bim and Bcl-2/Bim complexes to either ABT-263 treatment or BH3-2A mutations by increasing the local concentration of the proteins on mitochondria. However, instead of an uninterrupted hydrophobic sequence of more than 15 residues typical of tail-anchor proteins, the Bim CTS consists of two short hydrophobic sequences and includes multiple positively charged Arg residues, two of which are near the center of the sequence (*Figure 7A*, green).

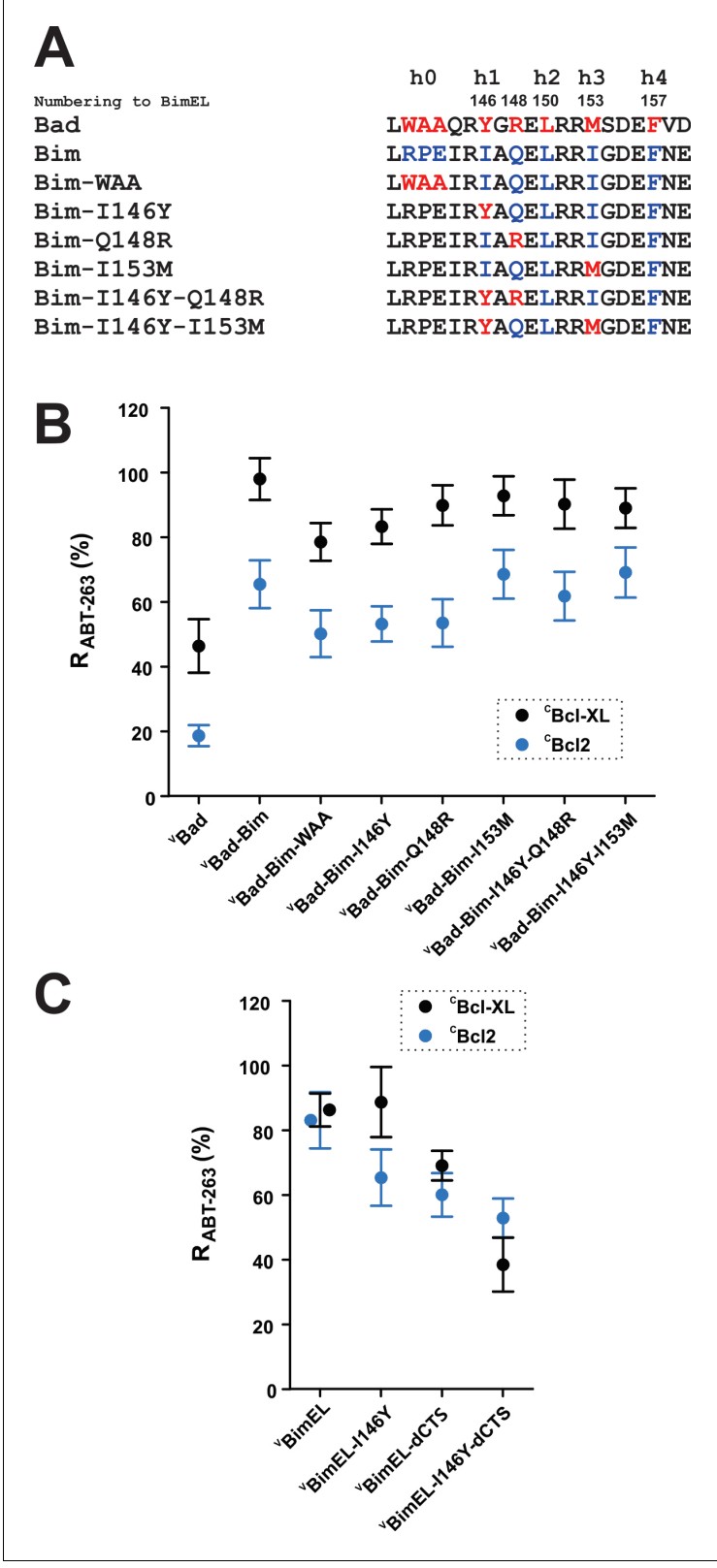

**Figure 6.** h0 and h1 residues in the Bim BH3 contribute to the resistance of Bcl-XL:Bim complexes to ABT-263. (**A**) Sequence alignment of the BH3 regions of Bad and Bim and the sequences of $^V$Bad-Bim mutants, residues from Bad (red), Bim (blue). (**B**) Identification of conserved hydrophobic residues in the Bim BH3 that contributed to $R_{ABT-263}$ for $^C$Bcl-XL:$^V$Bad-Bim (black) and $^C$Bcl-2:$^V$Bad-Bim (blue) complexes. The $R_{ABT-263}$ values of $^C$Bcl-XL:$^V$Bad-Bim

*Figure 6 continued on next page*

*Figure 6 continued*

and $^C$Bcl-2:$^V$Bad-Bim complexes from *Figure 4b* and *Table 1* were included to facilitate direct comparison. (**C**) The BH3 h1 residue I146 contributes to Bim binding to Bcl-2 and Bcl-XL. Substitution of I146 with the corresponding residue from Bad (BimEL-I146Y) decreased $R_{ABT-263}$ for $^C$Bcl-XL: $^V$BimEL/$^V$BimEL-dCTS (black) and $^C$Bcl-2:$^V$BimEL/$^V$BimEL-dCTS (blue) complexes. Data are mean ±95% confidence intervals calculated from FLIM-FRET binding curves shown in *Figure 6—figure supplement 1*.

DOI: https://doi.org/10.7554/eLife.37689.023

The following source data and figure supplements are available for figure 6:

**Source data 1.** Source data for the calculation of R values for mutants demonstrating that the h0 and h1 residues in the Bim BH3 contribute to the resistance of Bcl-XL:Bim complexes to ABT-263.

DOI: https://doi.org/10.7554/eLife.37689.024

**Figure supplement 1.** The h0 and h1 residues in the Bim BH3 region contribute to the resistance of Bcl-XL:Bim and Bcl-2:Bim complexes to ABT-263.

DOI: https://doi.org/10.7554/eLife.37689.025

**Figure supplement 1—source data 1.** Source data fitted to a Hill equation to determine the extent to which residues in the Bim BH3 region contribute to the resistance of Bcl-XL:Bimand Bcl-2:Bim complexes to ABT-263.

DOI: https://doi.org/10.7554/eLife.37689.026

---

To examine the importance of sequences within the Bim CTS for ABT-263 resistant binding of $^V$BimEL to $^C$Bcl-XL and $^C$Bcl-2, mutants were generated in which hydrophobic (I181, L185, I188 and V192) and hydrophilic (R186 and R190) residues of Bim were substituted with the negatively charged amino acid Glutimate (E) or the hydrophobic residue Alanine (CTS-2A), respectively (*Figure 7A*). The impact of these mutations on both the subcellular localization of $^V$BimEL at mitochondria (measured by the Pearson's r with MitoTracker) and $R_{ABT-263}$ values were quantified for binding to $^C$Bcl-XL and $^C$Bcl-2. Compared to $^V$BimEL (Pearson's r ~ 0.4) mitochondrial localization of all of the mutants was similarly poor (Pearson's r 0.1–0.3) (*Figure 7B–C*), and only slightly greater than the Pearson's r measured for the cytoplasmic Venus control. However, the mutants displayed different binding properties to the anti-apoptotic proteins (*Figure 7C* and *Figure 7—figure supplement 1*). For example, even though localization at mitochondria for the $^V$Bim-CTS2A mutant was impaired, it bound to both anti-apoptotic proteins in an ABT-263 resistant fashion ($R_{ABT-263}$ ~89%, CI 6% and 81%, CI 11%, for $^C$Bcl-XL and $^C$Bcl-2, respectively) similar to $^V$BimEL (85%, CI 5% and 83%, CI 9%; *Table 1* and *Figure 7C*). In contrast, $R_{ABT-263}$ for $^V$BimEL mutants L185E (42%, CI 10%), I188E (51%, CI 8%) and V192E (70%, CI 8%) exhibited gradually increasing binding to $^C$Bcl-XL (*Table 1*) without a corresponding increase in binding to mitochondria (*Figure 7C*). When all of the data were analyzed in aggregate using Spearman's rank-order correlation to include relationships that might not be linear the result was no correlation between localization for Bcl-XL and a weak correlation for Bcl-2. We suspect the latter may be due to the fact that Bcl-2 is constitutively membrane bound while Bcl-XL is located in both the cytoplasm and on the membrane. Thus in cells, localization to mitochondria and binding to anti-apoptotic proteins are independent functions of the Bim CTS.

The effect of the Bim CTS on binding to membranes and Bcl-XL was also assessed directly using purified proteins and liposome membranes. As above, binding of BimL and BimL-mutants to Bcl-XL was measured by FRET for incubations with and without liposomes. Displacement from Bcl-XL of the BimL mutants I181E, L185E, I188E and dCTS (numbered according to the BimEL sequence to facilitate comparison) were measured for different concentrations of ABT-263 (*Figure 7D*). Consistent with our observations in live cells, the binding of BimL-dCTS to Bcl-XL was much more sensitive to ABT-263 (IC50 ≈ 200 nM) compared to BimL (IC50 >>4 µM). These IC50 values are difficult to compare directly with $R_{ABT-263}$ values because the protein concentrations are different. Nevertheless, in this cell free assay using purified BimL and Bcl-XL, mutation of I181, L185 and L188 in the CTS region all resulted in increased sensitivity to displacement by ABT-263 (*Figure 7D*). As expected (*Pécot et al., 2016*), BimL binding to Bcl-XL in the absence of membranes was more easily displaced by ABT-263, but still required drug concentrations greater than 1 µM (compare *Figure 7D* with E). In solution, the mutations further decreased resistance of Bcl-XL/BimL complexes to ABT-263 (*Figure 7E*) confirming they augment binding of BimL to Bcl-XL independently of binding to membranes. The simplest explanation for ABT-263 resistant binding is a direct interaction between the Bim CTS and Bcl-XL in the context of the full length Bim protein.

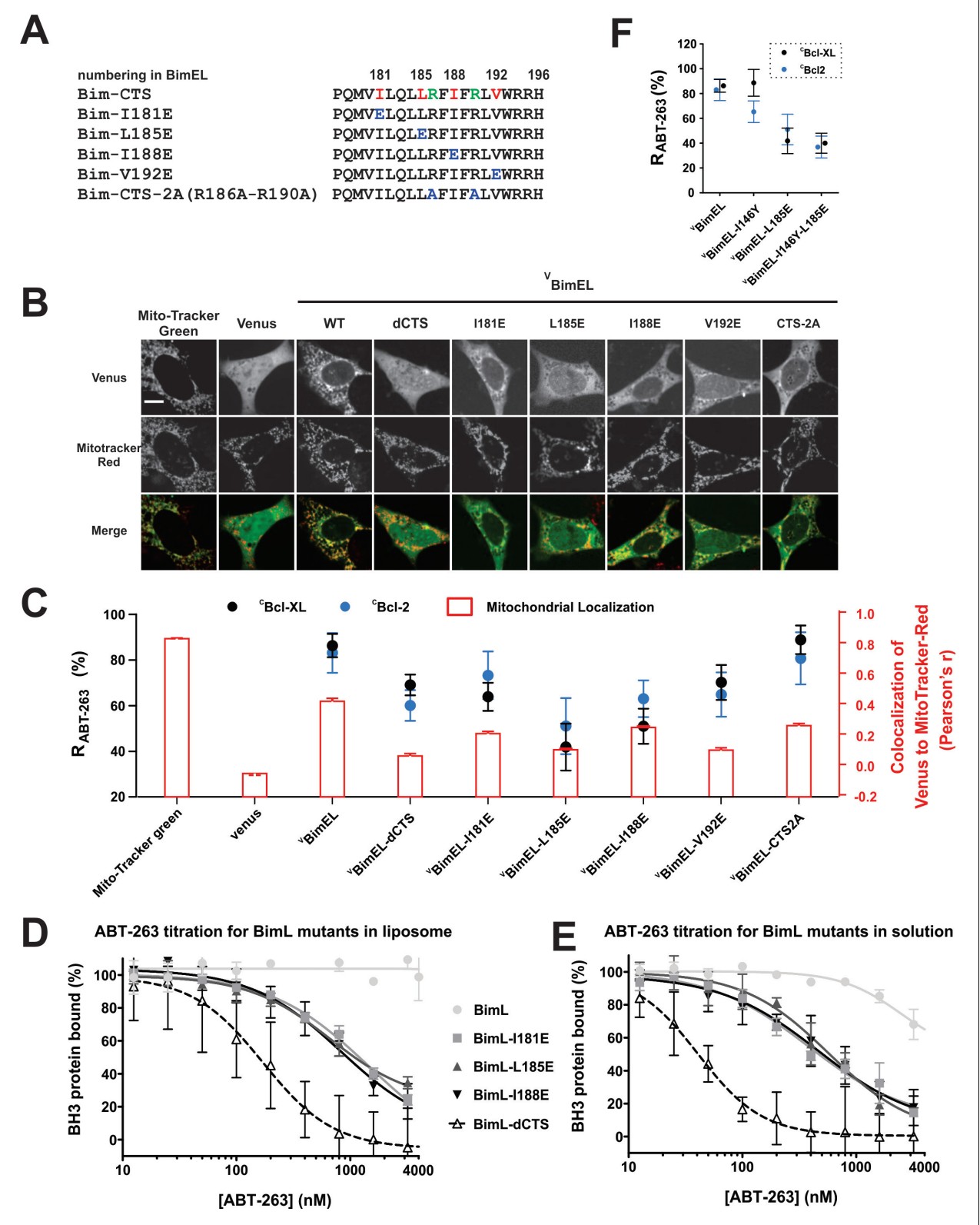

**Figure 7.** The Bim CTS binds to Bcl-XL and Bcl-2 independent of binding to membranes (**A**) Sequence alignment for the Bim CTS and mutants. Bim h1-h4, red; arginine residues within the CTS, green; substitutions, blue; numbering is for BimEL. (**B**) Selected images showing sub-cellular localization of <sup>V</sup>BimEL-mutants compared to MitoTracker. The scale bar represents 10 μm. (**C**) Bim binding to Bcl-XL and mitochondria by the Bim CTS are independent. Co-localization (Pearson's r) for Venus and MitoTracker-Red signals (Red bars), where average ±SEM was calculated using mean Pearson's

*Figure 7 continued on next page*

*Figure 7 continued*

r values determined in three independent replicates. At least 150 transiently transfected cells were analyzed across replicates. $R_{ABT-263}$ of $^C$Bcl-XL:$^V$BimEL-mutant complexes (black) or $^C$Bcl-2:$^V$BimEL-mutant complexes (blue). Data are mean ±95% confidence intervals calculated from FLIM-FRET binding curves shown in *Figure 7—figure supplement 1*. $^V$BimEL (from *Figure 1*) and $^V$Bim-CTS2A mutant complexes have similar high $R_{ABT-263}$ values although mitochondrial localization for $^V$Bim-CTS2A is impaired. $^V$BimEL-L185E, $^V$BimEL-I188E, $^V$BimEL-V192E and $^V$Bim-CTS2A are all poorly localized at mitochondria yet have increasing $R_{ABT-263}$ values. (D–E) BimL binding to Bcl-XL is improved by binding to membranes but most resistance to ABT-263 is due to Bim CTS dependent binding to Bcl-XL. Purified Bcl-XL and BimL protein binding in the presence of different concentrations of ABT-263 quantified in vitro using purified full-length proteins with (D) and without (E) liposomes. Control experiments demonstrating efficient binding of BimL to liposomes and Bcl-XL binding data for the Bim mutants are presented in the companion paper (*Chi et al., 2019*). Data are mean ±SD (n = 3 independent experiments). (F) Mutation of BimEL-L185 to E reduced $R_{ABT-263}$ for $^C$Bcl-XL:$^V$BimEL (black) and $^C$Bcl-2:$^V$BimEL (blue) complexes. However, additional mutation of I146 did not further reduce $R_{ABT-263}$. Data are mean ±95% confidence intervals calculated from FLIM-FRET binding curves shown in *Figure 7—figure supplement 1*.

DOI: https://doi.org/10.7554/eLife.37689.027

The following source data and figure supplements are available for figure 7:

**Source data 1.** Source data for the experiments demonstrating that the Bim CTS binds to Bcl-XL and Bcl-2 independent of binding to membranes.
DOI: https://doi.org/10.7554/eLife.37689.028

**Figure supplement 1.** Mutations in the Bim CTS reduce the affinity of binding to Bcl-XL and Bcl-2.
DOI: https://doi.org/10.7554/eLife.37689.029

**Figure supplement 1—source data 1.** Source data fitted to a Hill equation to quantify the effects of the indicated mutations in the BimCTS on binding affinities for Bcl-XL and Bcl-2.
DOI: https://doi.org/10.7554/eLife.37689.030

Mutation of the BH3 residue I146 to Y slightly exacerbated the effect of deletion of the entire CTS from Bcl-XL on $R_{ABT-263}$ (*Figure 6C*). However, when the I146Y mutation was combined with mutation of L185E we were unable to record a decrease in $R_{ABT-263}$ (*Figure 7F*) suggesting that the effects of the individual point mutants are not additive. Alternatively the effect size may be too small to measure by FLIM-FRET, a result consistent with the L185E mutation having less effect on $R_{ABT-263}$ than deletion of the CTS (*Table 1*).

## The Bim CTS interacts with Bcl-XL through hydrophobic residues

A direct interaction between the Bim CTS and Bcl-XL is incompatible with current assumptions that the Bim CTS inserts into lipid bilayers as a transmembrane helix similar to other tail-anchor proteins (*Petros et al., 2004*). Therefore, we expressed a version of BimEL with Venus fused to the C-terminus (BimEL$^V$) to prevent the Bim CTS from adopting a transmembrane topology. Distance constraints mean that this protein can only undergo FRET with Bcl-XL if the Venus protein is located on the cytoplasmic side of the membrane.

Fusion of Venus to the C-terminus of BimEL did not abolish localization of BimEL$^V$ at mitochondria (*Figure 8A and C*, red bars) and did not impair ABT-263 resistant binding to $^C$Bcl-XL (*Figure 8B–C*). Moreover, the FLIM-FRET efficiency observed was higher for BimEL$^V$ than for $^V$BimEL binding to $^C$Bcl-XL (*Figure 8B*) a result inconsistent with a transmembrane topology for the Bim CTS, as it suggests that Venus on the C-terminus of Bim is closer than Venus on the N-terminus of Bim to the mCer3 on the N-terminus of Bcl-XL. These results not only confirm the two independent functions of the Bim CTS, but also suggest that when bound to membranes the residues within the Bim CTS that bind $^C$Bcl-XL, for example L185, are on the cytoplasmic side of the membrane.

To examine the topography and Bcl-XL binding of the Bim CTS in solution and membrane bound states directly, we used a well-established procedure in which an environment sensitive fluorescent dye N,N0-dimethyl-N-(Iodoacetyl)-N0-(7-nitrobenz-2-oxa-1,3-diazol-4-yl) ethylenediamine (NBD) was attached to recombinant BimL at specific sites and the environment of the dye was measured by fluorescence spectroscopy (*Kale et al., 2014*). A series of mutants were generated in which individual residues across the Bim CTS were replaced with cysteine to enable NBD-labeling. As a solvent exposed control, a Bim mutant was prepared with a cysteine located at position 41. The fluorescence intensities of the dye labeled mutants were then recorded in the absence or presence of liposomes, Bcl-XL, and/or the aqueous quencher iodide.

The fluorescence of NBD increases when the dye inserts into a lipid bilayer or becomes deeply buried in the interior of a protein (*Johnson, 2005*). When Bim bound to Bcl-XL in solution there

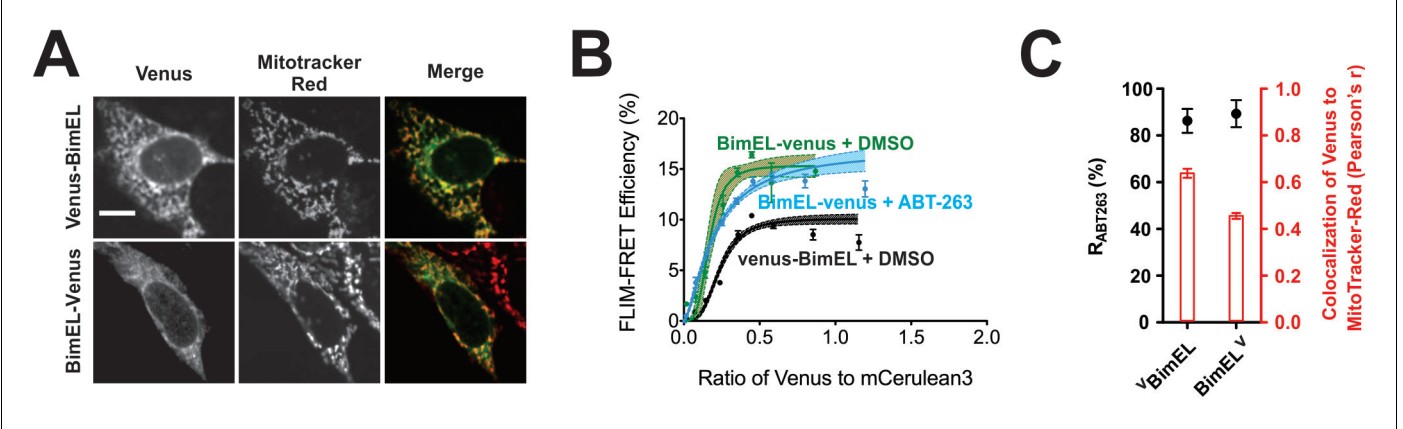

**Figure 8.** Bim CTS binds with non-transmembrane topology to cellular membranes. (A) Selected images for sub-cellular localization of [V]BimEL and BimEL[V] in BMK-DKO cells compared to MitoTracker. The scale bar represents 10 μm. (B) In live cells [C]Bcl-XL binding to BimEL[V] is resistant to ABT-263. FLIM-FRET binding curves for the interactions between [C]Bcl-XL and BimEL[V] in ABT-263 (blue) or DMSO (green) treated BMK-DKO cells expressing [C]Bcl-XL. Binding of [C]Bcl-XL to [V]BimEL (black) measured in the same three experiments is shown for comparison. Data from ROIs from three independent experiments were combined and used to generate binding curves with 95% confidence intervals as in *Figures 1* and *2*. (C) Sub-cellular localization of BimEL[V] is impaired compared to [V]BimEL but ABT-263 resistant binding to [C]Bcl-XL (R$_{ABT-263}$) is unchanged. Colocalization for Venus and MitoTracker-Red was measured in this experiment for manually selected regions of interest from transiently transfected cells expressing [V]BimEL or BimEL[V] (Pearson's r, red); Error bars, SEM, n > 30 cells. R$_{ABT-263}$ of [C]Bcl-XL:[V]BimEL or [C]Bcl-XL:BimEL[V] complexes, black dots. Data are mean ±95% confidence intervals calculated from FLIM-FRET binding curves shown in panel b.

DOI: https://doi.org/10.7554/eLife.37689.031

The following source data is available for figure 8:

**Source data 1.** Source data fitted to a Hill equation demonstrating that BimEL-venus undergoes FRET with mCer3-Bcl-XL.

DOI: https://doi.org/10.7554/eLife.37689.032

were no significant changes in hydrophobicity at any of the positions of the probe suggesting that during complex formation these residues do not become sufficiently buried to be protected from water (*Figure 9A*, diagrammed in 9C). In contrast, when BimL bound to liposomes the NBD fluorescence increased at positions 179–182 and 191–195 suggesting that these two regions anchor the CTS to the lipid bilayer (*Figure 9D*, diagrammed in 9F). Complex formation with Bcl-XL on the membrane did not markedly change the pattern of which amino acids increased in hydrophobicity, suggesting that complex formation does not change the way the Bim CTS interacts with membranes substantially (*Figure 9G* diagrammed in 9I).

Iodide quenching was used to identify residues protected by protein-protein interactions in addition to those protected by binding to membranes. Due to the relatively large size of iodide, residues involved in protein-protein interactions are often protected from quenching by iodide but unlike residues interacting with lipids the NBD fluorescence does not increase substantially because there is no protection from water (*Johnson, 2005*).

When Bim was incubated with Bcl-XL in solution, comparison with the control position 41 revealed that the NBD probes in the Bim CTS were not protected from iodide (*Figure 9B–C*, *Figure 9—figure supplement 1*), suggesting a relatively low affinity interaction between the Bim CTS and Bcl-XL in solution. As expected, when bound to membranes the residues at both ends of the Bim CTS that interact with the membrane were protected from iodide (*Figure 9E* compare residues 179–182 and 191–195). However, residues in the middle of the sequence surrounding residue 185 exhibited variable levels of protection from iodide with the NBD located at position 185 as accessible to iodide as the soluble control residue 41 (*Figure 9E*, *Figure 9—figure supplement 1*). Thus, in membrane bound Bim, the region between the two membrane binding sites in the Bim CTS remains on the surface of the membrane where it is accessible to iodide (diagrammed in *Figure 9F*). In contrast, when both Bcl-XL and liposomes were added the entire Bim CTS was protected from iodide (*Figure 9H*, *Figure 9—figure supplement 1*) strongly suggesting that on membranes Bcl-XL binds to or at least covers-up the central region of the Bim CTS (diagrammed in *Figure 9I*).

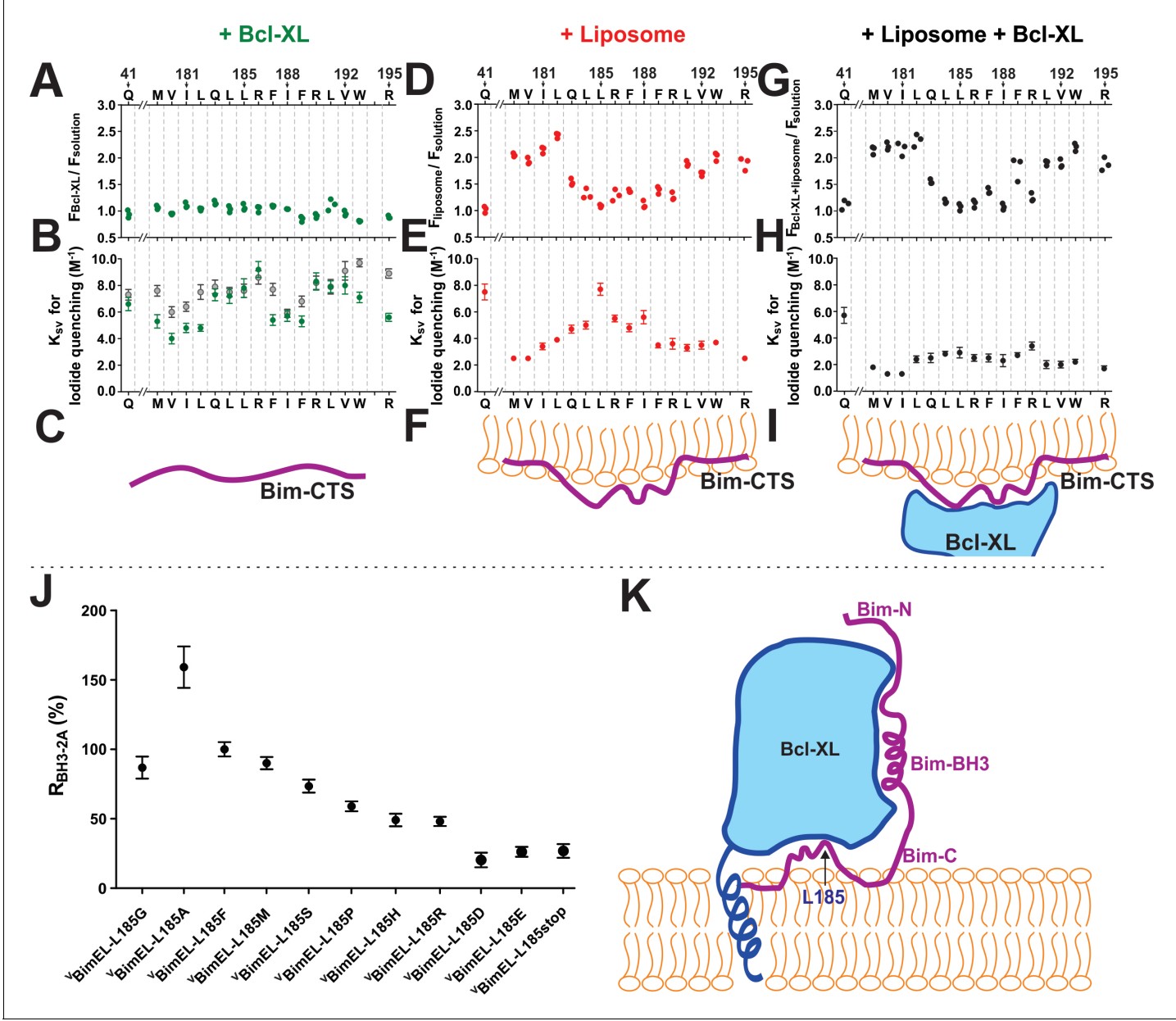

**Figure 9.** The Bim CTS binds to the cytoplasmic surface of membranes enabling concomitant binding to Bcl-XL (**A–I**) Interaction of the Bim CTS with liposomes and Bcl-XL measured using purified recombinant BimL protein. The amino-acids and positions above the panels indicate the residues in BimL exchanged for single-cysteines for labeling with NBD (numbering for BimEL, subtract 56 for BimL numbering). Top row, (**A,D,G**) NBD fluorescence changes in response to the addition of (**A**) Bcl-XL (green), (**D**) liposomes (red), (**G**) Bcl-XL and liposomes (black). Larger numbers indicate increased hydrophobicity of the environment of the NBD dye. Middle row, (**B,E,H**) iodide quenching constants ($K_{SV}$) for the same mutants and binding partners as above. Smaller $K_{SV}$ values indicate protection from iodide. Data are mean ±95% confidence intervals for linear fitting of the data as shown for exemplar raw quenching data, in **Figure 9—figure supplement 1**. Lower row, (**C,F,I**) illustrations of possible interactions of the Bim CTS based on the data above each. (**J**) Substitution of the L at position185 of BimEL with a charged residue abolishes resistance to ABT-263 while hydrophobic residues are tolerated. $R_{BH3-2A}$ for $^C$Bcl-XL:$^V$BimEL complexes from FLIM-FRET binding curves shown in **Figure 9—figure supplement 2** were calculated as in **Figure 2**. Data are mean ±95% confidence intervals for fit of the binding model. (**K**) Schematic model of the double-bolt locked Bcl-XL:Bim:membrane complex. Bcl-XL, cyan; Bim, purple; membrane, orange. The BH3 region of Bim engages the hydrophobic groove of Bcl-XL. The Bim CTS adopts a conformation in which the two ends bind to the membrane and the central region, particularly residue 185, binds to Bcl-XL. The C-terminus of Bcl-XL also interacts with the membrane (**Yao et al., 2015**).

DOI: https://doi.org/10.7554/eLife.37689.033

The following source data and figure supplements are available for figure 9:

**Source data 1.** Source data for Interaction of the Bim CTS with liposomes and Bcl-XL measured using purified recombinant full length proteins.

*Figure 9 continued on next page*

*Figure 9 continued*

DOI: https://doi.org/10.7554/eLife.37689.034

**Figure supplement 1.** Residues in the Bim CTS are protected from iodide when bound to membranes and Bcl-XL.

DOI: https://doi.org/10.7554/eLife.37689.035

**Figure supplement 1—source data 1.** Source data for Stern-Volmer quwnching plots for representative mutants of Bim.

DOI: https://doi.org/10.7554/eLife.37689.036

**Figure supplement 2.** The Bim CTS adopts a non-conventional conformation that allows binding to both membranes and Bcl-XL through hydrophobic interactions.

DOI: https://doi.org/10.7554/eLife.37689.037

**Figure supplement 2—source data 1.** Source data fitted to a Hill equation for the mutants illustrating that the Bim-CTS binds both to membranes and to Bcl-XL.

DOI: https://doi.org/10.7554/eLife.37689.038

As position L185 was shown to have the largest effect on binding of BimEL to both $^C$Bcl-XL and $^C$Bcl-2 (largest decrease in R$_{ABT-263}$, *Figure 7C*), and is central to the region protected from iodide when BimL bound to Bcl-XL (compare *Figure 9E and H*) we made a series of mutations at this location for $^V$BimEL-BH3-2A. In general, exchanging L185 with other amino acids with hydrophobic or polar non-charged side-chains (G, F, M, and S) did not affect the response of $^C$Bcl-XL:$^V$BimEL-mutant complexes to BH3-2A mutations significantly, except for the L185A mutant that displayed increased binding affinity (increased R$_{BH3-2A}$) (*Figure 9J* and *Figure 9—figure supplement 2*). For Bim-L185P, R$_{BH3-2A}$ decreased by about 20%, possibly due to a change in the structure or rigidity of the Bim CTS. However, replacing L185 with hydrophilic residues (H, R, D or E) decreased R$_{BH3-2A}$, by ~30–60%, comparable to truncation of the CTS at position 185 (L185stop) or deletion of the entire CTS region ($^V$BimEL-dCTS, *Figure 4C*). These results strongly suggest that L185 interacts with Bcl-XL through hydrophobic interactions.

Overall our data suggest a revised view of the Bcl-XL:Bim interaction (diagrammed in *Figure 9K*). Because Bim is bound to Bcl-XL by two different regions, the Bim BH3 and Bim CTS, the proteins are effectively double-bolt locked together.

## Discussion

Using FLIM-FRET and full-length proteins in live cells and in biochemical assays with mitochondria and liposomes, we discovered that the binding interactions between Bcl-XL and Bim are more extensive and result in a much higher affinity interaction than measured previously using peptides or protein fragments (*Chen et al., 2005*). Although previous results ascribed an increased apparent affinity to binding to membranes (*Pécot et al., 2016*) our results demonstrate that the increased binding affinity of Bim compared to Bad for Bcl-XL is primarily due to the combined effect of interactions with both the Bim BH3 region (residues 140–160) and the central hydrophobic region of the Bim CTS (residues 184–190). While our data also confirm that the interaction occurs optimally at the membrane, the increase in affinity due to binding to membranes is relatively small (*Figure 7D–E*). Together the two Bcl-XL binding sites in Bim confer resistance to the inhibitors ABT-263, A-1155463 and A-1331852 (*Figure 2D*) and to mutation of Bim BH3 residues at h2 and h4 that we confirmed as the major anti-apoptotic protein binding residues for both tBid and Bad in live cells (*Figure 1D–E*, and *Figure 2B–C*).

The importance of residues within the BH3-region of Bim other than those at the h2 and h4 positions in binding to Bcl-XL and Bcl-2 (*Figure 6*) was not detected in previous studies using BH3 peptides. Moreover, it was not possible to detect the Bcl-XL binding function of the Bim CTS using pull-down strategies, most likely because the required detergents disrupt hydrophobic interactions. Therefore, to determine the effect of mutations and drugs on interactions we used an automated system to collect the data required to generate large numbers of FLIM-FRET binding curves each requiring minimally hundreds, but typically made up of thousands of measurements from at least three independent experiments.

Substitution of candidate residues within the h0-h4 region of the Bim BH3 sequence with the ones in the Bad BH3 revealed that in live cells the h0 group WAA and h1 residue I146 contribute to the binding of Bim to both Bcl-XL and Bcl-2 (*Table 1*). Because the residues at h2 and h4 are the

same in Bad and Bim, the residues at h0 and h1 account for part of the increased affinity of binding of anti-apoptotic proteins by Bim compared to Bad.

In sum the additional interactions of the Bim BH3 region and the BIM CTS contributed roughly equally to the affinity of the interaction between Bim and Bcl-XL (*Figure 4B*, compare $^V$BimEL-Bad and $^V$BimEL-dCTS). Together deletion of the Bim CTS and mutation of h2 and h4 in the BH3 region of Bim almost eliminated binding to both Bcl-XL and Bcl-2 (*Figure 4C*). Moreover, the Bim CTS directly contributes to interactions with Bcl-XL (*Figure 5* and *Figure 7D–E*) and much of this interaction is due to residue L185 (*Figure 9*). Consistent with this interpretation weak crosslinking was observed between this region of Bim and Bcl-XL (*Chi et al., 2019*). Thus unlike conventional lock and key models, the interaction of Bim with Bcl-XL and Bcl-2 is more reminiscent of a double-bolt lock with binding mediated by two distant interacting surfaces (separated by ~40 residues) working together. The affinity of the CTS region alone for Bcl-XL is low as shown by replacing the BH3 region in Bim with a mutant BH3 region from Bad that does not bind Bcl-XL, (*Figure 4C*, BimEL-Bad, $R_{BH3-2A}$ ~26%). However, even in this construct it appears that the CTS contributes to binding as removing the CTS (BimEL-Bad-dCTS) further reduced binding ($R_{BH3-2A}$ ~10%). Moreover, double-bolt lock binding even with a reduced affinity BH3 sequence results in a very stable interaction (BimEL $R_{BH3-2A}$ ~80%). This mechanism also explains how Bim differs from Bad and tBid, which can be displaced from Bcl-XL and Bcl-2 by ABT-263. Unlike what was concluded from cell free assays using partial proteins, a double-bolt lock model predicts that in cells displacement of Bim from anti-apoptotic proteins will require novel strategies targeting both interacting sites.

Our re-evaluation of the Bim CTS not only challenges the conventional belief that the sequence only functions to bring and keep Bim at the outer mitochondrial membrane (*O'Connor et al., 1998*; *Terrones et al., 2008*) but suggests that the Bim CTS does not adopt the previously proposed trans-membrane topology (*Petros et al., 2004*). Instead, the CTS is pinned to the membrane by hydrophobic sequences at either end while the middle is exposed to the cytoplasmic environment (*Figure 9D–F*). We speculate that this topology allows the central region of the Bim CTS to bind to the membrane surface electrostatically via two Arg residues (*Figure 7B–C* and *Figure 7—figure supplement 1*). This hypothesis is consistent with our demonstration that substitution of these Arg residues with Ala decreased localization of the protein to membranes (*Figure 7B–C*) and may explain why there is no correlation between Bim mitochondrial localization and the translocase complex on the outer membrane despite colocalization (*Frank et al., 2015*). Binding of this region via hydrophobic residues at each end and electrostatically via arginine side chains is an arrangement that may facilitate Bim CTS binding to Bcl-XL through hydrophobic interactions including residue L185 (*Figure 9J*). This insight suggests the Bim CTS functions as an entirely new type of membrane binding domain. Moreover, the nature of the interactions with Bcl-XL and the observation that in healthy cells Bim is not bound to anti-apoptotic proteins suggest it may be possible to generate a pair of small molecules that selectively kill cancer cells by efficiently releasing Bim from or preventing Bim binding to Bcl-2 and Bcl-XL. However, given that Bim also activates both Bax and Bak and functional assays in cells demonstrating the importance of MCL-1 in MCF-7 cells (*Figure 3E*), our data strongly suggest that the effects of BH3 mimetics may be difficult to predict for different cell and tissue types.

## Materials and methods

**Key resources table**

| Reagent type (species) or resource | Designation | Source or reference | Identifiers | Additional information |
|---|---|---|---|---|
| Antibody | antibody to Bcl-XL (Rabbit polyclonal) | PMID: 14681679 | | (1:10,000), human, mouse reactivity |
| Antibody | antibody to Beta-actin (Mouse monoclonal) | Abgent | Cat. #: 8H10D10 | (1:5000), human, rat, mouse reactivity |

*Continued on next page*

*Continued*

| Reagent type (species) or resource | Designation | Source or reference | Identifiers | Additional information |
|---|---|---|---|---|
| Antibody | Donkey anti-rabbit (polyclonal) | Jackson Immuno Research Laboratories | Cat. #: 711-035-150 | (1:10,000) |
| Antibody | Donkey anti-mouse (polyclonal) | Jackson Immuno Research Laboratories | Cat. #: 711-035-152 | (1:10,000) |
| Cell line (*H.sapiens*) | MCF-7 | PMID: 3790748 | | Dr. Ronald N. Buick (University of Toronto) |
| Cell line (*M. musculus*) | Baby Mouse Kidney(BMK)-DKO (Bax and Bak knockout) cells | PMID: 11836241 | | Dr. Eileen White (Rutgers University) |
| Chemical compound, drug | Navitoclax; ABT-263 | Selleckchem | Cat. #: S1001 | in DMSO |
| Chemical compound, drug | A-1155463 | Chemietek | Cat #: CT-A115 | in DMSO |
| Chemical compound, drug | A-1331852 | Chemietek | Cat. #: CT-A115 | in DMSO |
| Chemical compound, drug | S-63845 | Chemietek | Cat. #: 1799633-27-4 | in DMSO |
| Chemical compound, drug | Venetoclax; ABT-199 | Chemietek | Cat. #: CT-A199 | in DMSO |
| Chemical compound, drug | TMRE | ThermoFisher Scientific | Cat. #: T669 | |
| Chemical compound, drug | DRAQ5 | Biostatus, UK | Cat. #: DR05500 | |
| Chemical compound, drug | MitoTracker Red | ThermoFisher Scientific | Cat. #: M22425 | |
| Chemical compound, drug | MitoTracker Green | ThermoFisher Scientific | Cat. #: M7514 | |
| Chemical compound, drug | Alexa 647-maleimide | ThermoFisher Scientific, Molecular probes | Cat. #: A20347 | |
| Chemical compound, drug | Alexa568-maleimide | ThermoFisher Scientific, Molecular probes | Cat. #. A20341 | |
| Chemical compound, drug | NBD | Molecular Probes | Cat. #: D-2004 | |
| Chemical compound, drug | PC (L-α-phosphatidylcholine) | Avanti Polar Lipids | Cat. #: 840051C | for making liposomes, used 48% PC |
| Chemical compound, drug | DOPS (1,2-dioleoyl-sn-glycero-3-phospho-L-serine) | Avanti Polar Lipids | Cat. #: 840035C | for making liposomes, used 10% DOPS |

*Continued on next page*

*Continued*

| Reagent type (species) or resource | Designation | Source or reference | Identifiers | Additional information |
|---|---|---|---|---|
| Chemical compound, drug | PI (L-α-phosphatidylinositol) | Avanti Polar Lipids | Cat. #: 840042C | for making liposomes, used 10% PI |
| Chemical compound, drug | PE (L-α-phosphatidyl ethanolamine) | Avanti Polar Lipids | Cat. #: 841118C | for making liposomes, used 28% PE |
| Chemical compound, drug | TOCL, (18:1 Cardiolipin) | Avanti Polar Lipids | Cat. #: 710335C | for making liposomes, used 4% TOCL |
| Commercial assay or kit | Fugene HD | Promega | Cat. #: E2311 | |
| Commercial assay or kit | TransIT-X2 | Mirus | Cat. #: Mir 6003 | |
| Gene (*H. sapiens*) | Bax | PMID: 14522999, | GI: L22473.1 | For recombinant protein |
| Gene (*H. sapiens*) | Bcl-XL | PMID: 18547146 | GI: Z23115.1 | For recombinant protein |
| Gene (*H. sapiens*) | Bcl-2 | PMID: 22464442 | GI: M14745.1 | For expression of $^C$Bcl-2 in cells |
| Gene (*H. sapiens*) | Bad | PMID: 22464442 | GI: AB451254.1 | For expression of $^V$Bad in cells |
| Gene (*H. sapiens*) | Bcl-XL | PMID: 22464442 | GI: NM_138578.3 | For expression of $^C$Bcl-XL in cells |
| Gene (*M. musculus*) | Bid | PMID: 16642033, PMID: 19062087 | GI: NM_007544.4 | For recombinant protein |
| Gene (*M. musculus*) | BimL | this paper | GI: AAD26594.1 | This lab, plasmid # 2187, for recombinant BimL purification |
| Gene (*M. musculus*) | tBid | PMID: 22464442 | GI: NM_007544.4 | for expression of $^V$tBid in cells |
| Gene (*M. musculus*) | BimEL | PMID: 22464442 | GI: XM_006498614.3 | for expression of $^V$BimEL in cells |
| Other | Cell Carrier-384, Ultra | PerkinElmer | Cat. #: 6057300 | for live cell imaging |
| Other | Non-binding surface, 96-well plate, black with clear bottom | Corning | Cat. #: 3881 | For recombinant protein and liposome assays critical to use non-binding plate |
| Other | Opera Phenix | PerkinElmer | Cat. #: HH14000000 | |
| Other | ISS-Alba | PMID: 25631031 | | Custom built by ISS for DWA lab |
| Software, algorithm | GraphPad Prism | San Diego, California | Version 6 | Scientific graphing program, used to perform statistical analysis |
| Software, algorithm | ImageJ | PMID: 17936939 | MBF - ImageJ for microscopy, Dr. Tony Collins (McMaster University) | FLIM-FRET analysis Macro, in this paper: https://github.com/DWALab/Liu_et_al_2018_eLife |

*Continued on next page*

*Continued*

| Reagent type (species) or resource | Designation | Source or reference | Identifiers | Additional information |
|---|---|---|---|---|
| Software, algorithm | CellProfiler | PMID: 17269487 | | Colocalization analysis, in this paper: https://github.com/DWALab/Liu_et_al_2018_eLife |
| Transfected Construct | mVenus-pEGFP-C1 | other | GI: KU341334.1 | Dr. Ray Truant (McMaster University). Backbone EGFP-C1 (Clonetech) |
| Transfected Construct | mCerulean3-pEGFP-C1 | PMID: 21479270 | | Dr. Mark A Rizzo (University of Maryland). Backbone EGFP-C1 (Clonetech) |
| Transfected Construct | pSPUTK | Stratagene Santa Clara CA | Cat. #: CB4278654 | Cotransfected to reduce overexpression in live cells |

## Constructs and compounds for assays in live cells

Plasmids encoding the mCerulean3 (mCer3), or Venus fluorescence proteins in place of the EGFP coding region in pEGFP-C1 (Clontech) were kind gifts from Mark A. Rizzo (University of Maryland) and Ray Truant (McMaster University) respectively. To generate plasmids encoding the fusion proteins the required coding regions were amplified by PCR and inserted into the plasmids encoding the appropriate fluorescence protein. All of the plasmids included coding sequences for a GGS linker sequence between the coding regions except for $^V$Bad (linker sequence, SGLRSRGG) and BimEL$^V$ (linker sequence, SRGGGPVAT). All the swapping and site-directed mutants were obtained through PCR-based mutagenesis using oligonucleotides from Integrated DNA Technologies and Phusion DNA polymerase from New England Biolabs. ABT-263 was purchased from Selleckchem, A-1155463,A-1331852 and ABT-199 were from Chemietek.

## Cells, culture conditions and transfections

The MCF-7 human breast cancer cell line was cultured in Dulbecco's Modified Eagle Medium (DMEM, ThermoFisher) supplemented with 10% of fetal bovine serum (HyClone) and non-essential amino acids (NEAA, ThermoFisher). Transfections were performed using Fugene HD reagent according to the manufacturer's standard protocol (Promega). MCF-7 cell clones stably transfected with vectors encoding mCer3, $^C$Bcl-XL or $^C$Bcl-2 were selected in DMEM supplemented with 10% fetal bovine serum, non-essential amino acids (NEAA, ThermoFisher) and 500 µg/ml neomycin. After 3 weeks colonies were isolated and cultured as above. The baby mouse kidney (BMK) cell line in which the genes for Bax and Bak have been deleted (BMK-DKO) was a kind gift from the originator Eileen White and was cultured as above. The two originating cell lines used in the studies reported here (MCF-7 and Baby Mouse Kidney cells with both Bax and Bak knocked out) and all stably transfected clones were shown to be free of mycoplasma using a PCR based test. The MCF-7 cells were authenticated by the hospital for Sick Children facility in Toronto. Clones that stably express mCer3 or $^C$Bcl-XL were selected and cultured with 5 µg/mL Blasticidin S. For microscopy, cells were seeded in 384-well imaging plates (PerkinElmer) and cultured as above for 24 hr prior to transient transfection. Cells treated with ABT-263, A-1155463 or A-1331852 were incubated at 37°C for 18–24 hr in a fresh media containing 20 µM drug before imaging unless indicated otherwise.

## Image based cell death assay

MCF-7 cells grown in a 384 well plate (Cell Carrier Ultra) were transfected, using the TransIT-X2 (Mirus) reagent and manufacturer's protocol, with plasmids encoding Venus protein alone, or Venus protein fused to BimEL ($^V$BimEL), tBid ($^V$tBid) or Bad ($^V$Bad). Media was changed 3 hr later, and cells were incubated 24 hr. Prior to imaging, cells were stained with 10 nM tetramethylrhodamine (TMRE, to stain mitochondria transmembrane potential, Life Technologies) and 5 µM DRAQ 5 (to stain

nucleic acids, Biostatus, UK) and incubated for at least 30 mins at 37°C, 5% $CO_2$. Micrographs of the cells were acquired on an OPERA Phenix (PerkinElmer), using the 20x water immersion lens (NA 1.0). Data were acquired within 2 hr after staining and during imaging the cells were maintained at 37°C, 5% $CO_2$. Four channels were collected simultaneously: mCer3 (Ex 425 nm, Em 435–480 nm), Venus (Ex 488 nm, Em 500–550 nm), TMRE (Ex 561, Em 570–630), DRAQ 5 (Ex 640 nm, Em 650–760 nm). Acquisition was automated and acquisition settings were identical between wells and for each independent replicate (n = 3). Acquisition settings were different for *Figures 3* and *5* therefore, the X-axis is not comparable. For the data in *Figure 5* the cloning vector, pSPUTK (Stratagene Santa Clara CA) was used to dilute the plasmid encoding the $^V$BH3 proteins, to reduce overexpression of Venus-tagged protein while ensuring consistent transfection efficiencies.

Cell images recorded on the OPERA Phenix were imported into Harmony high-content analysis software (PerkinElmer). The seven step pipeline built to identify live and dead cells was as follows: 1) Nuclear and total cell areas for each cell were identified using DRAQ five intensity data and a Harmony segmentation algorithm. 2) The cytoplasmic area was obtained by subtracting the nuclear area from the total cell area. 3) Cell images that extended to or beyond the any edge of the micrograph were removed from the analysis as the total area is uncertain. 4) The total intensity of TMRE within each cell image was calculated. 5) Nuclear area and morphology features (e.g. roundness) were calculated. 6) Cell area and morphology features were calculated. As positive and negative controls, >100 cells treated with 500 ng/ml TNFα and cycloheximide (apoptotic cells) and >100 untreated cells, respectively were analyzed similarly for each experiment. 7) The intensity of Venus and mCerulean3 was measured for all selected cells. Data from the control cells were used to train a linear classifier in the Harmony software. Each cell death curve represents measurements for at least 1000 cells for each of three replicates.

To confirm that the linear classifier reported data similar to manual analysis a subset of the data was reanalyzed with manual thresholds to score cells as, 'TMRE negative' (TMRE intensity at least two standard deviations lower than the average in an untreated well) or 'small nuclear area' (Nuclear area at least two standard deviations lower than the average in an untreated well).

Cell death was also measured as a function of the amount of the transiently expressed protein, such as $^V$BimEL, in cells. For this analysis individual cells were classified as alive or dead and then results were binned by Venus intensity. For these experiments the Venus intensity serves as an estimate of the amount of the BH3 protein produced in the cell. The range of intensities within each bin was kept consistent for each transfectant, for all three independent replicates. Mean % Dead and standard error was plotted for each intensity bin.

## Fluorescence lifetime measurements

Steady-state fluorescence images and fluorescence lifetime images were both acquired using a custom-built Alba confocal microscope from ISS (Champagne, Illinois), which measures fluorescence lifetime by time correlated single photon counting (TCSPC). Channels of mCer3 and Venus were acquired simultaneously using 445 nm and 514 nm pulse interleaved excitation (PIE) through a 442/512/561 multiband filter and emission split by a 520nm-longpass filter was collected through 459–499 nm and 528–555 nm bandpass filters, respectively. All images were acquired as 256 pixels x 256 pixels at 25°C using a 60 × 1.3 NA PlanApo water immersion objective and laser powers set to minimize photobleaching and photon pileup during acquisition with 0.1 ms per pixel dwell time and 5-frame repeat. TCSPC-data were processed using VistaVision software (ISS) and the average lifetime for each pixel was obtained by fitting FLIM pixels binned to ensure a total photon count >1000 for each set of binned pixels analyzed. The lifetime and photon count for each pixel in both channels were exported for additional analysis using ImageJ.

## Measuring protein:protein interactions in cells by FLIM-FRET

For Bcl-2 family proteins the problem of measuring protein:protein interactions in live cells (*Herce et al., 2013*) is particularly acute as the proteins often function differently at membranes therefore, assays using cell extracts do not capture either the correct or the complete set of activities (*Lovell et al., 2008*). Moreover, solubilizing membranes requires detergents artificially increase and decrease Bcl-2 family protein:protein interactions (*Lovell et al., 2008*). The use of purified proteins and cell free assays has provided much useful information, however, due to the difficulty of purifying

full-length Bcl-2 family proteins, many assays in vitro are based on truncated proteins or peptides and the environment does not well represent that of an intact cell.

We used FLIM-FRET in live cells using fluorescent protein tagged full-length proteins to quantitatively study protein:protein interactions for Bcl-2 family proteins and to assess the impact of small-molecule inhibitors on these interactions (*Aranovich et al., 2012*; *Kale et al., 2012*; *Liu et al., 2012*). This technique removes the disadvantages of biochemical methods, but has its own limitations. For example the over-expression of a fusion protein with a fluorescent protein may result in protein concentrations within cells that are above the Kds for the interactions, affecting the relative Kds measured. For these reasons it is difficult to compare interactions between different pairs of proteins, instead the method is best suited to comparing the effects of small molecules and mutations on interactions.

In our application of FLIM-FRET the ratio of Venus to mCer3 intensities was used as an estimate of the relative amounts of the proteins to plot binding curves. Stable expression of mCer3 and using measurements from cells with little variation in amount of mCer3 allows ratio intensities to serve as a reasonable approximation for generating binding curves, but distorts the meaning of the Hill coefficients obtained. Moreover, in some cells high expression of Venus reduces the expression of mCer3 and for the some of the proteins analyzed this effect can dominate at high ratios of Venus to mCer3. As a result, in some experiments the data at ratios of Venus to mCer3 greater than 0.5 are not well described by a standard binding curve (e.g. *Figure 2—figure supplement 1*, panel C, black). For this reason quantitative comparisons were restricted to those regions of the curves that are fit using a standard binding curve with a Hill slope. For $^V$tBid and $^V$Bad it was not possible to measure binding to $^C$Bcl-XL at ratios of Venus to mCer3 intensities greater than ~0.7 in the presence of ABT-263 because the released $^V$BH3-proteins killed the cells.

Another complication with using any kind of FRET in live cells is the impact of collisions that can generate false positive signals. Fortunately, the latter can be distinguished by measuring binding curves using FLIM-FRET. Collisions vary directly with protein concentration and therefore generate straight lines rather than curves that saturate (e.g. *Figure 4—figure supplement 1E*, red).

To limit the effect of compounds or mutagenesis potentially changing both the binding affinity and the maximum FRET efficiency to different extents and still generate a single descriptor that can be used to describe the interactions, we introduced the concept of Resistance (R) of a complex to a specific compound ($R_{ABT-263}$) or mutation ($R_{BH3-2A}$) (*Figure 2A*). Quantitative analysis based on values of R allows comparison between multiple samples each that results in a unique binding curve and that was generated from measurements of thousands of different intracellular ROIs that would be hard to compare otherwise. R values cannot replace the original binding curves for detailed determination of how the interaction is disturbed. Binding curves for $^C$Bcl-XL and $^V$Bad-Bim-I146Y with and without the BH3-2A mutation allowed quantification of the decrease in binding affinity due to the I146Y mutation (*Figure 6*).

To generate binding curves, ROIs containing mitochondria were selected, and the corresponding intensities in both mCer3 and Venus channels and the average lifetime for mCer3 were extracted from the original TCSPC data using a customized ImageJ script. ROIs were automatically discarded from the analysis if the $^C$BclXL signal was too low and therefore noisy, or close to saturated, or the Venus signal was close to saturated. The ratio of Venus to mCer3 was calculated using the intensities of each channel. FLIM-FRET efficiency (E%) was calculated for each ROI as: $E\% = (1-\tau i/\tau 0)\times 100\%$, where $\tau i$ is the mean lifetime for that ROI and $\tau 0$ is the average lifetime for mCer3 for all ROIs not expressing detectable Venus. FLIM-FRET efficiency was then distributed into bins of the same size according to Venus:mCer3 intensity ratio (bin size 0.1 for ratios lower than 0.5, 0.25 for ratios between 0.5 and 1.25, 0.5 for ratios higher than 1.25) and plotted (±se) against Venus:mCer3 ratio. The binding curves were fitted using GraphPad Prism version 5.0d for Macintosh (GraphPad Software, San Diego California USA, www.graphpad.com) with the function: $E\% = Emax \times (I_{Venus}\div I_{mCerulean3})^h/[Kd^h+(I_{Venus}\div I_{mCerulean3})^h]$. Emax is the maximum FLIM-FRET efficiency corresponding to saturation of donor binding sites by an acceptor; $I_{mCerulean3}$ and $I_{Venus}$ are intensities of mCer3 and Venus, respectively; Kd is the relative equilibrium dissociation constant; h is the Hill slope.

## Cellular localization analysis

BMK-DKO cells transiently expressing Venus-tagged constructs of interest were stained with 5 μM DRAQ 5 (BioStatus) to stain nucleic acids, and 500 nM MitoTracker-Red to stain mitochondria, (Life Technologies) in DMEM at 37°C and 5% $CO_2$, 30 min. One positive control well was additionally stained with 500 nM MitoTracker-Green (to stain mitochondria, Life Technologies). Cells were incubated at 37°C, 5% $CO_2$ during sample imaging. For each well, 60 images at 256 × 256 pixels resolution were recorded using a 40x water immersion (NA 1.1) lens on an Opera Phenix confocal microscope. DRAQ 5 (Ex 640 nm, Em 650–760 nm), Venus (Ex 488 nm, Em 500–550 nm) and Mito-Tracker-Red (Ex 561 nm Em 570–630 nm) channels were acquired sequentially, automatically switching between channels at each field of view.

An analysis pipeline was created in Cell Profiler 2.2.0. In this pipeline, the DRAQ five channel was used to identify nuclei, and total cell area with smoothing to ensure the entire cell is selected as one region. Cells that extend to the border of the image were rejected from, the analysis. The cytoplasm for each cell was identified as an object (Cytoplasm = 'Total cell area' – 'nuclear area'). Objects suitable for analysis were identified by mean Venus intensity to identify cells expressing appropriate levels of the Venus-tagged proteins of interest. A median Venus intensity (low threshold) was used to discard any cells that passed the mean Venus filter due to improper segmentation (ie. overlap between an untransfected cell and a bright transfected cell). Objects were further selected by mean MitoTracker-Red intensity (high and low threshold), to ensure appropriate staining. A size filter was used to remove small dead cells or debris and any abnormally large (possibly improperly segmented) cells. Background was subtracted for each image and a Pearson's correlation coefficient (Pearson's r) between Venus and MitoTracker-Red signals was calculated for all remaining objects. In each replicate, a minimum of 20 cells were identified per protein of interest and mean Pearson's r was calculated. In sum, at least 150 cells were analyzed per sample. The average and standard error of the mean Pearson's r values determined for the three replicates was plotted using GraphPad Prism.

## Recombinant protein purification and labeling

Full length and single cysteine mutants of Bcl-XL and tBid were purified as described previously (*Kale et al., 2014*). For Bim, the cDNA encoding full-length wild-type murine BimL was introduced into pBluescript II KS(+) vector (Stratagene, Santa Clara CA). Sequences encoding a polyhistidine tag followed by a TEV protease recognition site (MHHHHHHGGSGGTGGSENLYFQGT) were added to create an in-frame fusion to the N-terminus of BimL. BimL-dCTS was constructed by introducing a stop codon (TAA) after position P121 to delete the entire CTS. Single cysteine mutants were generated by PCR-based mutagenesis using oligonucleotides from Integrated DNA Technologies and Phusion DNA polymerase from New England Biolabs.

The recombinant proteins were expressed in Arabinose Induced (AI) Escherichia coli strain (Life Tech, Carlsbad, CA). *E. coli* cells were lysed by mechanical disruption with a French press. The cell lysate was separated on a Nickel-NTA column (Qiagen, Valencia CA) to bind the recombinant His-tag fused proteins and after washing a buffer containing 300 mM imidazole was applied to elute the proteins. This elution was then adjusted to 150 mM NaCl and applied to a High Performance Phenyl Sepharose (HPPS) column. BimL was eluted with a no salt buffer and dialyzed against a buffer containing 10 mM HEPES pH7.0, 20% Glycerol, and then flash-frozen and stored at −80°C.

Single cysteine mutants of Bcl-XL and tBid were labeled with the indicated maleimide-linked fluorescent dyes as described previously (*Kale et al., 2014*; *Lovell et al., 2008*). Single cysteine mutants of BimL were labeled with the same protocol as tBid with the exception that the labeling buffer also contained 4M urea.

## FRET measurements of interactions between recombinant proteins

Single cysteine mutants of BimL (41C) and tBid (126C) were purified and labeled with Alexa 568-maleimide. A single cysteine mutant of Bcl-XL (152C) was purified and labeled with Alexa 647-maleimide. Alexa568 labeled BimL or tBid was incubated with either Alexa647-labeled or as a control unlabeled Bcl-XL along with the indicated concentrations of ABT-263. The intensity of Alexa568 fluorescence with unlabeled or Alexa647-labeled Bcl-XL was measured as $F_{unlabeled}$ or $F_{labeled}$ respectively. FRET, indicating protein-protein interaction, was quantified using the decrease of Alexa568 fluorescence when BimL or tBid bound to Alexa647-labeled Bcl-XL compared to unlabeled Bcl-XL.

FRET efficiency was calculated as: FRET efficiency (%)= $(1-F_{labeled}/F_{unlabeled})*100\%$ as described previously for cBid-Bcl-XL. Binding was measured after incubation at 37°C for 1 hr.

To compare the inhibitory effect of ABT-263 on different BimL mutants or tBid binding to Bcl-XL, ABT-263 was titrated into a solution containing 10 nM BH3 protein (BimL or tBid) and 40 nM Bcl-XL. At these concentrations BH3 protein binding to Bcl-XL had saturated (BH3 protein bound = 100%) and represents the maximum FRET efficiency ($F_{max}$). Inhibition of BH3 protein binding to Bcl-XL by ABT-263 was measured based on the decrease of FRET (BH3 protein bound = $F/F_{max}*100\%$). Data were fitted to a standard inhibitor dose response equation with a Hill slope using GraphPad Prism (V6.02). For BH3 proteins in which binding was poorly inhibited (BimL (*Figure 2E* and *Figure 7D–E*), BH3 protein bound was assumed to be 0% at an infinite concentration of ABT-263. Loss of protein on the cuvettes and other surfaces at low concentrations required use of protein at concentrations higher than the Kds of binding. For this reason the dissociation constant is not independent of protein concentration and the data in *Figure 2E*, *Figure 7D–E* report relative displacements comparable between figures rather than absolute affinity.

### NBD fluorescence assay and iodide quenching

The single cysteine mutants of Bim were purified and labeled with NBD (N,N0-dimethyl-N-(Iodoacetyl)-N0-(7-nitrobenz-2-oxa-1,3-diazol-4-yl) ethylenediamine; Molecular Probes, Cat. #: D-2004). The NBD fluorescence assay and iodide quenching of NBD-labeled Bim mutants was performed as described previously for Bax (*Kale et al., 2014*), with the exception that the assays were performed in 100 ul volume in low protein binding 96-well plates and fluorescence was measured using a plate reader (Tecan M1000). In the experiments reported here, 20 nM NBD-labeled Bim alone or 20 nM Bim plus 50 nM of Bcl-XL were incubated with 0.2 mg/mL (final lipid concentration) of liposomes. For the 'In solution' control, the protein(s) were incubated with an equal volume of assay buffer instead of liposomes.

## Acknowledgements

Jarkko Ylanko provided invaluable assistance in running the Opera and analyzing the data generated. Nehad Hirmiz developed the Cell Profiler Script used to analyze the colocalization data. Funding was provided by grant FDN 143312 from the Canadian Institutes of Health Research. DWA holds the Tier 1 Canada Research Chair in Membrane Biogenesis.

## Additional information

### Funding

| Funder | Grant reference number | Author |
|---|---|---|
| Canadian Institutes of Health Research | FDN143312 | David William Andrews |
| Canada Research Chairs | Tier 1 | David William Andrews |

The funders had no role in study design, data collection and interpretation, or the decision to submit the work for publication.

### Author contributions

Qian Liu, Conceptualization, Software, Formal analysis, Supervision, Funding acquisition, Validation, Investigation, Methodology, Writing—original draft, Writing—review and editing; Elizabeth J Osterlund, Data curation, Formal analysis, Validation, Investigation, Writing—review and editing; Xiaoke Chi, Software, Formal analysis, Validation, Investigation, Methodology, Writing—review and editing; Justin Pogmore, Formal analysis, Investigation, Methodology, Writing—review and editing; Brian Leber, Conceptualization, Formal analysis, Supervision, Funding acquisition, Investigation, Writing—original draft; David William Andrews, Conceptualization, Formal analysis, Supervision, Funding acquisition, Methodology, Writing—original draft, Writing—review and editing

Author ORCIDs

Xiaoke Chi (iD) https://orcid.org/0000-0002-5269-8389

David William Andrews (iD) http://orcid.org/0000-0002-9266-7157

Decision letter and Author response

Decision letter https://doi.org/10.7554/eLife.37689.041

Author response https://doi.org/10.7554/eLife.37689.042

## Additional files

### Supplementary files

• Transparent reporting form

DOI: https://doi.org/10.7554/eLife.37689.039

### Data availability

Data analysed during this study are included in the manuscript and supporting files. Source data files have been provided for Figures and most of the supplements. Software scripts are available at Github (https://github.com/DWALab/Liu_et_al_2018_eLife; copy archived at https://github.com/elifesciences-publications/Liu_et_al_2018_eLife) and www.andrewslab.ca.

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
