## [Decision Letter]

Thank you for submitting your article "Bim escapes displacement by BH3-mimetic anti-cancer drugs by double-bolt locking both Bcl-XL and Bcl-2" for consideration by *eLife*. Your article has been reviewed by three peer reviewers, including Volker Dötsch as the Reviewing Editor and Reviewer #1, and the evaluation has been overseen by Philip Cole as the Senior Editor.

The reviewers have discussed the reviews with one another and the Reviewing Editor has drafted this decision to help you prepare a revised submission.

Summary:

The manuscript by Andrews and colleagues applies a series of sophisticated in-cell protein interaction monitoring analyses, with correlative biochemical studies, to evaluate the susceptibility of BH3-only/anti-apoptotic complexes to pharmacologic dissociation by small molecule inhibitors. The conclusion is that complexes involving BIM are stronger owing to a previously unappreciated secondary binding interface between the BIM C-terminus and anti-apoptotic Bcl-XL/Bcl-2. Although some of the data are not as definitive as the expressed interpretations, the general concepts are reasonably well demonstrated and the capacity to conduct such studies in live cells to better vet the state of these medically-relevant interactions is certainly appreciated.

Essential revisions:

1) The authors should demonstrate a functional relationship between the observed differences in protein complex dissociation and phenotypic outcome (i.e. apoptotic cell death). The recommended trio of treatments is ABT-263 (dual inhibitor), a selective Bcl-XL inhibitor and, importantly, the selective Bcl-2 inhibitor ABT-199. The latter agent is the most clinically-relevant (the only FDA-approved drug) and should be included in the study. The two selective inhibitors would also serve as ideal controls for each other and better validate the specificity of this live-cell protein interaction system.

2) The authors need to demonstrate the functional relevance of the double-bolt lock binding concept in cells. Although the authors indicate that the double-bolt lock mechanism has "profound implications for the utility of BH3-mimetics as drugs," the manuscript lacks the experimentation to support this conclusion. A straightforward experiment to address this key point would be to transiently express and compare the influence of wild-type, BH3 mutant, CTS mutant (e.g. a loss-of-function L185 mutation), and a double mutant on apoptosis induction/cell death and relative resistance to the anti-apoptotic inhibitors. Does eliminating the CTS component of the BimEL binding interaction increase the sensitivity to the anti-apoptotic inhibitors of Bcl-2 and Bcl-XL as the title/article suggests? What is the relative functional contribution of the BH3 and CTS binding components?

3) Measuring the release of cationic lipophilic dyes in many cases do not reflect MOMP or cell death. For instance, cultured cells can survive in the presence of protonophores for days in culture. Has cell death been evaluated with other measurements?

4) Figure 4—figure supplement 1, Figure 7—figure supplement 1, Figure 6—figure supplement 1 and Figure 9—figure supplement 2: for the various black/blue/red/green curves, it is unclear why the endpoints are variable (with respect to the x axis). As an example, it's not clear why sometimes the curves end at <1.0 on the x axis (as in the Bcl-2 column of Figure 6—figure supplement 3J-L), but other times not (Bcl-XL column of Figure 6—figure supplement 1D-F). This variation occurs within an individual subpanel and across datasets as well. If this relates to cell death due to transient BH3 protein expression and drug treatment, how do such differences in cell death induction under the various conditions influence the analyses?

5) Figure 2: for the constructs depicted in part A, is the CTS for Bad and Bad-Bim the native BAD CTS or the BIM CTS? Presumably it's the former, but based on the similar coloring it can be misinterpreted as the latter. For this figure, the data for all of the constructs depicted in A, should be plotted in parts B and C to allow for careful comparison (i.e. add back BimEL, BimEL-d20, Bad, Bad-Bim). Why does replacing the BAD BH3 sequence with BIM BH3 sequence restore vBad-Bim/Bcl-XL binding to BimEL levels? This suggests that the BIM CTS is not needed for the complete rescue. The R% value for vBad-Bim/Bcl-XL has a large error bar, making it unclear whether the vBimEL-Bad-dCTS construct fully abrogates Bcl-XL interaction upon ABT-263 treatment (as proposed) or not; this is an important experimental condition, so the result for this condition should be repeated/clarified.

6) Figure 4: with respect to dissociating the membrane targeting vs. Bcl-XL binding functionalities of the BIM-CTS, the authors indicate that, in Figure 4C, "the mitochondrial localization of all of the mutants was similarly poor" and conclude that the differences seen in R% are unrelated to membrane binding. However, the data in 4C could be equally interpreted (based on inspection alone) as showing a reasonable trend between mitochondrial localization and influence on R%, particularly since the data range is relatively narrow (40-70%). The authors' interpretation requires further statistical consideration. The potential similarities between the data in 4D with regard to aqueous vs. liposomal conditions would be very appealing (and supportive of the authors' conclusion regarding the BIM-CTS), although the authors do not demonstrate that BimL is membrane-localized in the liposomal condition. The presence and extent of added BimL at the liposomal membrane should be documented (e.g. western blot of aqueous phase vs. liposomal pellet), since it is important to determine whether the two panels are truly reflecting the different conditions of aqueous BimL vs. liposomal BimL.

7) Mutational analysis was performed only on the BH3 proteins, not on Bcl-2 or Bcl-XL. Usually, binding models are confirmed by mutations on both binding partners and amino acid exchanges. This would also identify binding sites on the Bcl-2 family proteins.

---

## [Author Response]

Summary:The manuscript by Andrews and colleagues applies a series of sophisticated in-cell protein interaction monitoring analyses, with correlative biochemical studies, to evaluate the susceptibility of BH3-only/anti-apoptotic complexes to pharmacologic dissociation by small molecule inhibitors. The conclusion is that complexes involving BIM are stronger owing to a previously unappreciated secondary binding interface between the BIM C-terminus and anti-apoptotic Bcl-XL/Bcl-2. Although some of the data are not as definitive as the expressed interpretations, the general concepts are reasonably well demonstrated and the capacity to conduct such studies in live cells to better vet the state of these medically-relevant interactions is certainly appreciated.

We greatly appreciate the positive comments from the reviewers and the thoughtful, constructive ideas/suggestions made. In response we have attempted to address each of these as thoroughly as possible. As a result not only has this manuscript been significantly improved by the addition of new functional data, a number of additional changes have been made that improve the clarity of the way the data is presented.

Figure 3, Figure 5 and Figure 2—figure supplement 3 report new data. Additional images illustrating that the morphology changes do not alter our results have been added in response to a reviewer query. To confirm that a similar range of morphologies is present in all the data representative images of the FLIM-FRET data were also added to Figures 1, 2 and Figure 2 supplements. The colocalization data in Figure 7C was replaced with new data and analysis with a Cell Profiler pipeline to permit analysis for a larger number of images.

To improve the presentation and take advantage of the *eLife* format some figures were split (e.g. the original Figure 1 is now Figures 1 and 2) and the figure supplements for Figure 2 and Figure 9 were broken up into individual supplements. In the methods section, what was previously called "Cell viability and membrane potential measurements" has been replaced by “Image based cell death assay” because the membrane potential measurements made previously have been replaced with analyses using a validated trained linear classifier that integrates cell and nuclear morphology and TMRE data to classify cells as alive or dead/dying.

To address the functional consequences in live cells of the new binding site we discovered in Bim we provide a better description of the original data and have uploaded a new paper by Chi et al., entitled “The carboxyl-terminal sequence of Bim enables Bax activation and killing of unprimed cells” that we have submitted for potential co- or linked publication.

To generate a more easily viewed version of the required comparison between the original and the resubmitted manuscripts the figures were removed from the previous version prior to generating the comparison.

Finally, the software scripts have been uploaded to Github including a Powerpoint based description for the colocalization analysis.

Essential revisions:1) The authors should demonstrate a functional relationship between the observed differences in protein complex dissociation and phenotypic outcome (i.e. apoptotic cell death). The recommended trio of treatments is ABT-263 (dual inhibitor), a selective Bcl-XL inhibitor and, importantly, the selective Bcl-2 inhibitor ABT-199. The latter agent is the most clinically-relevant (the only FDA-approved drug) and should be included in the study. The two selective inhibitors would also serve as ideal controls for each other and better validate the specificity of this live-cell protein interaction system.2) The authors need to demonstrate the functional relevance of the double-bolt lock binding concept in cells. Although the authors indicate that the double-bolt lock mechanism has "profound implications for the utility of BH3-mimetics as drugs," the manuscript lacks the experimentation to support this conclusion. A straightforward experiment to address this key point would be to transiently express and compare the influence of wild-type, BH3 mutant, CTS mutant (e.g. a loss-of-function L185 mutation), and a double mutant on apoptosis induction/cell death and relative resistance to the anti-apoptotic inhibitors.

While we agree with the reviewer that this sort of data would be of significant interest addressing it is much more complicated than it appears. The result predicted from point 1 can be seen in Figure 1 in which in the presence of ABT-263 MCF-7 cells overexpressing Bcl-XL tolerate higher relative levels of BimEL expression than that of Bad or tBid (the truncation in the cyan curves). In the new version we have further clarified this in the text. However, for reasons outlined below this is difficult to quantify and due to the vagaries of transient transfection not obvious in all of the data in the manuscript. The trio of experiments shown in Figure 1 served as controls for many of the other experiments and have been repeated many times therefore we are very certain of our interpretation for this data. As described in detail in response to point 3, ABT-199 is fluorescent preventing FLIM FRET analysis (this now stated in the subsection “The impact of BH3 sequence mutations and small-molecule inhibitors on the interactions between anti-apoptotic proteins and BH3 only pro-apoptotic proteins”, fifth paragraph). Nevertheless, the functional data added for venetoclax in the new Figure 3D suggests the major inhibitor of apoptosis in MCF-7 cells is MCL-1 (Figure 3D-E) confounding use of venetoclax to demonstrate direct relevance of the double-bolt lock in live cells.

As this is an important issue, we have also added data to address the functional relationships to several figures. The basic idea is that BimEL should be more active than BimEL-dCTS to inhibit Bcl-XL and Bc-2 since the latter is lacking the second binding site. However, BimEL-dCTS efficiently inhibits MCL-1 and the MCF7 cells used for all of our experiments are highly dependent on MCL-1 for survival (new Figure 3E). Indeed we now show that the only inhibitor that killed MCF-7 cells efficiently was the MCL-1 inhibitor. The inhibitors to Bcl-2 and/or Bcl-XL resulted in minimal kill that was fully suppressed by expression of exogenous Bcl-XL or Bcl-2 (new Figure 3D-E). Therefore, while we demonstrated that the ^C^Bcl-XL and ^C^Bcl-2 constructs protect against TNFα and cycloheximide (well established to induce apoptosis), the protection offered by both anti-apoptotic proteins is equivalent in response to expression of BimEL and BimEL-dCTS (new Figure 5A). This result is due in part to the induction of cell death due to the Bim-BH3 binding to MCL-1 (new Figure 3E). To circumvent this problem we replaced the BH3 region in both Bim and Bim-dCTS with that of Bad. These hybrid molecules retain efficient binding to Bcl-XL and Bcl-2 but no longer bind Mcl-1. When the hybrid proteins are expressed in MCF-7 cells, the full length protein is now more efficient at killing MCF-7 cells than the version lacking the CTS (new Figure 5B) as predicted by the reviewers. These new results are presented in a subsection entitled “The CTS of Bim contributes to pro-apoptotic activity”. As described however, the results are complicated because the CTS of Bim not only increased the affinity of Bim for Bcl-XL and Bcl-2 but enables Bim to efficiently activate Bax. This function of the Bim CTS is addressed in cell lines and primary cells and the molecular mechanism is delineated in our new paper by Chi et al. that we have submitted for co-publication.

Thus, the only way we can address the biological effects of deletion of the CTS directly is using purified proteins. Therefore, we have also added data showing that in incubations with mitochondria deletion of the Bim CTS reduces the affinity for binding to Bcl-XL (new Figure 5C). However, as we point out the measured affinity is well below the expression level in cells. As a result the expectation would be that even at endogenous levels of Bim, BimEL-dCTS would bind and inhibit Bcl-XL in cells. In addition, we show that without the CTS, Bim is less effective at displacing tBid and activated Bax from Bcl-XL (new Figure 5D-E). Therefore, the biological outcome in cells of releasing or not releasing Bim from Bcl-2 and Bcl-XL is further complicated by both the activity of the anti-apoptotic protein and the functions of its binding partners including the released protein.

The same complications exist for the mutants that vary in resistance to the inhibitors of Bcl-XL and Bcl-2. The mutations that reduce binding affinity for Bcl-XL lead to a loss in activation of Bax and therefore, the data generated are difficult to interpret. In the new co-submitted manuscript by Chi et al. we examine the effects of the new binding site in Bim in multiple cell lines as suggested by the reviewer. We find that the different dependencies of different cell lines on the various anti-apoptotic proteins compared to susceptibility to activation of Bax means that the results of these kinds of experiments are at present unpredictable. Indeed we state in the attached new manuscript that the response of primary cortical neurons changes over time in culture (Chi et al.). It is for these reasons that we have been careful in the text of this paper to always state that the main consequence of the second binding site in Bim that we characterize is to make the effects likely to be obtained with small molecule BH3-mimetics unpredictable. We have now repeated that statement in the Discussion and feel that the data shown in the two papers more than adequately demonstrates this.

Does eliminating the CTS component of the BimEL binding interaction increase the sensitivity to the anti-apoptotic inhibitors of Bcl-2 and Bcl-XL as the title/article suggests?

The title of the paper indicates only that the second binding site enables Bim to not be displaced from Bcl-XL and Bcl-2 in cancer cells. We show that binding is resistant for both proteins for the main inhibitors of multi-domain anti-apoptotic proteins Bcl-XL and Bcl-2 and that Bim CTS mutants are displaced by the drugs in live cells. The one exception is ABT-199 (Venetoclax) which the reviewers correctly point out is the only inhibitor in clinical use. Although ABT-263 is in human clinical trials, ABT-199 is of obvious interest and importance and we have attempted to measure the effect of ABT-199 on displacement of BH3 proteins from Bcl-2 using FLIM-FRET. The difficulty is that ABT-199 is fluorescent. This is now stated in the fifth paragraph of the subsection “The impact of BH3 sequence mutations and small-molecule inhibitors on the interactions between anti-apoptotic proteins and BH3 only pro-apoptotic proteins”. Moreover, the fluorescence of ABT-199 is spectrally broad and it can undergo FRET with mCer3 tagged Bcl-2 (see data below). As a result Bcl-2 proteins not bound to a BH3 protein bind ABT-199 and there is a corresponding change in donor lifetime due to drug binding. Since we need to fit a biding curve to unambiguously differentiate collisions from authentic binding we need to measure FLIM at sub-saturating concentrations of the acceptor BH3-protein. As the amount of BH3-protein goes down (low ratios of venus to cerulean) the FRET between ABT-199 and Bcl-2 increases. Fitting a triple exponential model plays havoc with the analysis as reported by others (Garcia et al., 2007; Sun et al., 2011) and in the attempted experiments we obtained apparent dose response curves for ABT-199 ‘displacement’ of BH3 proteins that made no sense and did not fit a Hill equation at all. Unfortunately that means that FLIM FRET is not useful for measuring the effects of ABT-199. To get around this problem we have been attempting to generate a far-red pair of fluorescence proteins for FLIM use but to date the fluorescence lifetime of the donor proteins is too short to make reliable measurements of binding (Canty et al., 2018).

**Author response image 1. respfig1:** ABT199 significantly affects the lifetime of ^C^Bcl2. BMKd3 cells expressing ^C^Bcl2 were incubated 24 hours in DMEM media alone (untreated) or media plus DMSO, ABT199 or ABT263. FLIM data was collected on the ISS-Alba, each point represents average lifetime of 10 images collected per replicate (n=3). A one-way ANOVA test was performed in GraphPad Prism, with a Dunnett's Multiple Comparison post-test to compare to the untreated control (*** is p value < 0.0001).

As described above, eliminating the CTS or introducing point mutations in the CTS have unpredictable consequences in cells whether drug treated or not depending on the extent to which changes in inhibition of not just Bcl-2 and Bcl-XL but also Mcl-1 and activation of Bax impact cell viability. The issue is further complicated by differences in the extent of loading of Bcl-2 and Bcl-XL in the cell lines. Thus, the only way that we can address this interesting point is in vitro using purified proteins. Data demonstrating that mutations in the BimCTS increase the sensitivity to ABT-263 in vitro was included in the original manuscript (new Figure 7D-E). Thus, we have established in live cells that the extra binding site in Bim that we have discovered can have significant yet unpredictable effects on sensitivity and resistance of cells to different pro-apoptotic stimuli.

What is the relative functional contribution of the BH3 and CTS binding components?

This is another excellent question. In the new manuscript attached (Chi et al. submitted) we have measured the contribution of the BH3 and CTS binding components to Bax and Bcl-XL quantitatively and report the changes in Kd associated with each for binding to Bax. We are unable to measure changes in the affinity for Bcl-XL because all of the affinities are so high that the Kds are less than 1 nM. Thus the Kds for binding Bcl-XL are so low that the mutations in the CTS will not affect binding to Bcl-XL in a biologically relevant way unless a BH3 mimetic is added or another BH3 protein competes for binding. (new Figure 3C-E). Moreover, as shown in the new manuscript (Chi et al. submitted) we demonstrate that at physiological concentrations in cells, BOTH the BH3 and CTS sequences are necessary for BimEL to activate Bax. As predicted from the Kd measurements the BH3 region alone is sufficient to inhibit Bcl-XL, Bcl-2 and MCl^-^1 in multiple live cells. Therefore, Bim-dCTS kills cells that are ‘primed’ i.e. dependent on the expression of anti-apoptotic proteins because it can bind and inhibit all of the anti-apoptotic proteins. But full length Bim including the CTS is required to kill ‘unprimed’ cells (Chi et al.).

3) Measuring the release of cationic lipophilic dyes in many cases do not reflect MOMP or cell death. For instance, cultured cells can survive in the presence of protonophores for days in culture. Has cell death been evaluated with other measurements?

We agree with the reviewers that the release of cationic lipophilic dyes is not always sufficient to infer MOMP or cell death. However, when loss of mitochondrial transmembrane potential is the result of activation of Bax or Bak (as in the experiments reported here) cells do not recover and eventually die. Nevertheless, release of cationic lipophilic dyes is not the most accurate way to measure impending cell death. For this reason we have replaced loss of TMRE with a cell death score based on a multiparametric linear classifier based on nuclear condensation, cell shrinkage and loss of TMRE staining (described in the first paragraph of the subsection “Regulation of apoptosis by fluorescent fusion proteins in live cells” and Figure 3A). The classifier was trained on cells treated with TNFα and cycloheximide, a well-known method for induction of apoptosis. As all three events are common in cells undergoing apoptosis we feel that it is reasonable to conclude that the classifier identifies cells that have undergone or are undergoing apoptosis. We have explained this process in detail in the Materials and methods and provide references to the use of this approach with MCF-7 cells, other cell lines and primary cells. Our experience is that any cell classified as dead using this assay is committed to dying and will not survive even if the death stimulus is removed. The sensitivity of detection of apoptosis was improved using this approach as can be seen in the new cell death assays data presented in new Figures 3 and 5.

4) Figure 4—figure supplement 1, Figure 7—figure supplement 1, Figure 6—figure supplement 1 and Figure 9—figure supplement 2: for the various black/blue/red/green curves, it is unclear why the endpoints are variable (with respect to the x axis). As an example, it's not clear why sometimes the curves end at <1.0 on the x axis (as in the Bcl-2 column of Figure 6—figure supplement 3J-L), but other times not (Bcl-XL column of Figure 6—figure supplement 1D-F). This variation occurs within an individual subpanel and across datasets as well. If this relates to cell death due to transient BH3 protein expression and drug treatment, how do such differences in cell death induction under the various conditions influence the analyses?

There is variation in the amount of each acceptor protein that is produced in cells. Even though we have transfected with the same ratio of DNA to reagent, the amount of protein produced varies from construct to construct and between transfections. We assume that the amount of fluorescent mutant-protein generated depends on many factors including protein size, rate of folding, rate of degradation and whether that protein (or protein + treatment) kills the cell. As stated in the original manuscript one diagnostic of toxicity resulting in lower intensity levels (variable endpoints) is that the control curves will extend to higher ratios of venus to cerulean (mCer3) than when ABT-263 is added. This data remains in Figure 1 where we show that in the presence of ABT-263 the cell will tolerate production of more Bim than tBid or Bad. While it is tempting to ascribe this to the double bolt lock preventing displacement of BimEL (as a response to questions 1-2 above) the complications we describe above prevent us from doing so.

The question raised about how differences in cell death induction influence the analysis is a good one. We apologize for not being clear about our explicit controls in the original version of the manuscript. We stated only that we used a minimum and maximum threshold on the mCer3 (donor) channel prior to binning and plotting these curves. The minimum threshold ensures the Regions of Interest (ROIs) have sufficient counts for fitting the data properly. The maximum threshold excludes dead cells, because a condensed cell can have much higher counts (due to condensation and increased autofluorescence), than the average in a stable cell line. To demonstrate the morphology of the cells analyzed we have added images to Figure 1 and figure supplements for Figure 2. To ensure that we used appropriate thresholds we examined the data manually. We have now provided some sample data of the type we used to ensure that we had set the thresholds appropriately (described in paragraph four of subsection “The impact of BH3 sequence mutations and small-molecule inhibitors on the interactions between anti-apoptotic proteins and BH3 only pro-apoptotic proteins”, data shown in new Figure 2—figure supplement 3). These images have numbered ROIs and a corresponding table of lifetime measurements to demonstrate that we get the same range of values in cells with relatively normal and dying morphologies. Hopefully together with the expanded explanation in the methods section this will answer the question to the satisfaction of the reviewer. We appreciate the opportunity to provide this kind of information regarding our method that would typically be very difficult for us to publish. It is a real advantage to *eLife* as a journal.

5) Figure 2: for the constructs depicted in part A, is the CTS for Bad and Bad-Bim the native BAD CTS or the BIM CTS? Presumably it's the former, but based on the similar coloring it can be misinterpreted as the latter.

We apologize for the confusion. The supposition by the reviewer is correct. We have changed the coloring in the figure (now Figure 4) to make it more clear what the composition is of the different mutants.

For this figure, the data for all of the constructs depicted in A, should be plotted in parts B and C to allow for careful comparison (i.e. add back BimEL, BimEL-d20, Bad, Bad-Bim).

This was a point of discussion amongst the authors for the original manuscript. We agree that the data is more easily interpretable when all on the same axes and that the graphical display is easier to interpret than the table of all of the data. Therefore we modified the figure (now Figure 4) as requested. However, to ensure that we have not misrepresented it as new data we have added an explanatory note to the figure caption.

Why does replacing the BAD BH3 sequence with BIM BH3 sequence restore vBad-Bim/Bcl-XL binding to BimEL levels? This suggests that the BIM CTS is not needed for the complete rescue.

It is not clear why replacing the BAD BH3 with the BIM BH3 generates a mutant that resisted displacement. Similar to Bim both BAD and PUMA have unusual membrane binding domains at the C-terminus therefore it is possible that the BAD CTS contributes to the interaction in some way. We plan to examine this in more detail sometime in the future. It suggests that there may indeed be more than one sequence that can increase binding to Bcl-XL. However, the BAD BH3 does not bind to Bcl-XL tightly enough for the BAD CTS to confer resistance to BH3 mimetics. Resistance therefore, requires the BIM BH3 and the BAD CTS. This will be an interesting area for future investigation. We have addressed this observation in the subsection “Multiple residues in the Bim BH3 contribute to ABT-263 resistance”.

The R% value for vBad-Bim/Bcl-XL has a large error bar, making it unclear whether the vBimEL-Bad-dCTS construct fully abrogates Bcl-XL interaction upon ABT-263 treatment (as proposed) or not; this is an important experimental condition, so the result for this condition should be repeated/clarified.

We agree completely with the reviewer. The large error bar was due to a relatively small number of transfected cells that were analyzed due to low transfection efficiency with that plasmid. The experiment was repeated and the transfections were more successful and the number of images acquired was increased greatly increasing the number of cells that could be analyzed. As a result the error bar is much smaller now (now in Figure 4).

6) Figure 4: with respect to dissociating the membrane targeting vs. Bcl-XL binding functionalities of the BIM-CTS, the authors indicate that, in Figure 4C, "the mitochondrial localization of all of the mutants was similarly poor" and conclude that the differences seen in R% are unrelated to membrane binding. However, the data in 4C could be equally interpreted (based on inspection alone) as showing a reasonable trend between mitochondrial localization and influence on R%, particularly since the data range is relatively narrow (40-70%). The authors' interpretation requires further statistical consideration.

The extent to which membrane binding contributes to the increased affinity is an interesting question that we have put much thought into and attempted to address experimentally. In the original manuscript the Pearson’s correlation values were measured for multiple cells using manually defined regions of interest. In an attempt to make the data less subjective and to reduce the size of the error bars new images were acquired and used for the automated assessment of Pearson’s R (now in Figure 7). The algorithm used is included in CellProfiler Analyst. All of the correlations decreased due to inclusion of larger cytoplasmic areas yet to our surprise the answer obtained was almost identical even though more than ten times as many cells were analyzed. Visual inspection suggested to us that the mutants are similarly impaired for targeting to mitochondria (mitotracker) with co-localizations between 0.1 and 0.3. In contrast the R% seemed more variable particularly comparing residues 185, 188, 192. However, as requested by the reviewer all of the data was analyzed in aggregate using Spearmans rank-order correlation to include relationships that might not be linear. The result was no correlation between localization for Bcl-XL and a weak correlation for Bcl-2. We suspect the latter may be due to the fact that Bcl-2 is constitutively membrane bound while Bcl-XL is located in both the cytoplasm and on the membrane. We now report this data and statistical analysis in the second paragraph of the subsection “The Bim CTS increased Bim binding to Bcl-XL and Bcl-2 independent of Bim binding membranes”.

The potential similarities between the data in 4D with regard to aqueous vs. liposomal conditions would be very appealing (and supportive of the authors' conclusion regarding the BIM-CTS), although the authors do not demonstrate that BimL is membrane-localized in the liposomal condition. The presence and extent of added BimL at the liposomal membrane should be documented (e.g. western blot of aqueous phase vs. liposomal pellet), since it is important to determine whether the two panels are truly reflecting the different conditions of aqueous BimL vs. liposomal BimL.

We agree that this is an important control for the experiment presented and one that we had already performed in setting up the assay. We did not include this data in the original manuscript because it was generated as a standard part of optimizing the assay. The experiments showing efficient binding to liposomes of BimL and the differences in efficiencies for the mutants are included the new paper (Chi et al., submitted, copy included for reviewers) because they complete the table of binding constants in that paper and are important for showing that CTS binding to liposomes does not require the residues involved in activating Bax. That BimL binds efficiently to liposomes and that mutants also bind liposomes is now referred to in the current paper in the caption for the new Figure 7.

7) Mutational analysis was performed only on the BH3 proteins, not on Bcl-2 or Bcl-XL. Usually, binding models are confirmed by mutations on both binding partners and amino acid exchanges. This would also identify binding sites on the Bcl-2 family proteins.

While we agree this data would be satisfying it doesn’t impact the major observation that the individual residues we identify in CTS of Bim are responsible for resistance to BH3 mimetics and that the mechanism is increased binding affinity for anti-apoptotic proteins.

The mutagenesis experiments required to find the new binding site in Bim took 3 years. Mutants that do not fold properly are uninformative but take a lot of time to rule out and can cause one to miss the real binding site. NMR experiments by us and others were unsuccessful due to peak broadening when Bim or Bim peptides were added to Bax. There are no crystal structures that include the membrane binding domains of any of the Bcl-2 family proteins. Without any guide mutagenesis of Bcl-2 or Bcl-XL is not feasible. As a result we have been using crosslinking to attempt to find the binding site. This approach is giving good results for Bax and weak crosslinks to Bcl-XL (see data in new paper by Chi et al. submitted). However, determining the precise site is still a long way off. The crosslinks we have obtained to Bcl-XL are not efficient enough to pursue currently. Thus, it is likely we will have to infer the location on Bcl-XL from the site that we eventually identify for Bax. Therefore, identifying the binding site on the multi-domain anti-apoptotic proteins will have to remain a future endeavor.